# Achieve Latency-Efficient Tempora-Coding Spiking LLMs via Discretization-Aware Conversion

## Abstract

Large language models (LLMs) have achieved remarkable success while introducing critical energy bottlenecks that challenge sustainable deployment. Spiking neural networks (SNNs) provide a promising approach for energy-efficient spiking LLMs via ANN-to-SNN (A2S) conversion. Among various spike coding methods, time-to-first-spike (TTFS) coding is particularly appealing as it conveys information with a single spike, further reducing energy consumption. However, existing TTFS-based A2S conversion relies on continuous-time assumptions, requiring prohibitively large latencies (e.g., 4096 time steps) to approximate ANN's continuous values. This dependency leads to unacceptable inference delay in deep models, particularly LLMs, posing significant challenges for developing practical temporal-coding spiking LLMs. In this paper, we propose a discretization-aware theoretical framework that establishes a precise correspondence between discrete TTFS-based SNNs and ANNs. Our key insight reveals that conversion errors are bounded by latency-dependent terms. Motivated by these, we introduce the Quantization-Consistent ANN-to-SNN (QC-A2S) conversion, which integrates low-bit quantization with discretization-compatible TTFS neurons, achieving latency-efficient temporal-coding spiking LLMs. Comprehensive evaluation on LLaMA models demonstrates comparable performance with dramatically reduced latency.

## 1 Introduction

Large Language Models (LLMs) represent a paradigm shift in artificial intelligence, leveraging deep learning architectures trained on massive text corpora to capture intricate linguistic patterns, syntactic structures, and semantic relationships, thereby achieving remarkable capabilities in natural language understanding and generation (Zhang et al., 2022; Touvron et al., 2023; Achiam et al., 2023; Dubey et al., 2024). Most LLMs are built upon the Transformer architecture, which relies heavily on multi-head attention mechanisms and dense matrix multiplications, resulting in cubic computational complexity and substantial energy consumption during both training and inference (Vaswani et al., 2017; Zhao et al., 2023). Moreover, following the "scaling law", LLMs have grown from billions to trillions of parameters to achieve better performance, which further increases computational and storage demands (Chen et al., 2024a; Hoffmann et al., 2022). Consequently, the critical challenge facing the LLM community is developing approaches to reduce computational complexity and energy consumption while preserving model performance capabilities.

Spiking Neural Networks (SNNs) are biologically plausible computational models inspired by the mechanisms of neurons and synapses in the human brain (Maass, 1997; Roy et al., 2019). SNNs transmit and compute information asynchronously through discrete spike events rather than continuous-valued activation functions, demonstrating remarkable energy efficiency when implemented on specialized neuromorphic hardware (Yao et al., 2023; Zhou et al., 2022; Davies et al., 2018; Merolla et al., 2014). Consequently, developing **spiking LLMs** has emerged as a promising solution to address the substantial energy consumption challenges of LLMs. Currently, two primary approaches are used to develop spiking LLMs: direct training methods that incorporate surrogate gradients to address non-differentiability (Yao et al., 2023; Mukhoty et al., 2023; Zhou et al., 2024), and ANN-to-SNN (A2S) conversion methods that transfer pre-trained weights while preserving ap-

proximate equivalence through carefully designed techniques (Jiang et al., 2024; Chen et al., 2025a; Hao et al., 2023). Given the enormous computational and storage requirements of direct training for LLMs, practical spiking LLMs are predominantly achieved through A2S conversion for energy-efficient intelligent applications in resource-constrained environments (Xing et al., 2024a).

Beyond the rate coding commonly used in A2S conversion methods, recent neuroscience research has highlighted temporal-based spike coding that offer superiors energy efficiency advantages (Park et al., 2019; Zhang et al., 2019; Stanojevic et al., 2024). Temporal coding represents continuous values through precise spike timing rather than spike counts, suggesting that the representation of information depends on when the spikes occur (Gütig & Sompolinsky, 2006). Among various temporal codings, **time-to-first-spike (TTFS) coding** is particularly noteworthy, as it encodes information in the latency of a single spike, which substantially reduces energy consumption by minimizing spike counts (Park et al., 2020; Rueckauer & Liu, 2018).

Existing TTFS-based conversion methods underlying rely on continuous-time assumptions that directly approximate the continuous values of ANNs (Zhao et al., 2025; Stanojevic et al., 2024). However, practical hardware implementations impose discrete timing constraints through finite latency and clock granularity. Such discretization inevitably introduces conversion errors that severely compromise model accuracy. To mitigate the discretization-induced errors, existing methods require prohibitively large latency (e.g., 4096 time steps),

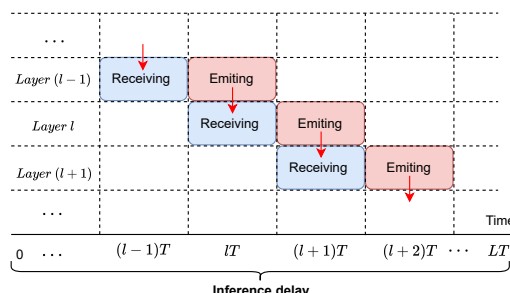

Figure 1: Inference delay across network layers.

causing extensive spike latency accumulation across network layers (Figure 1). This creates unacceptable inference delay in deep models, particularly for LLMs, posing significant challenges for developing practical **temporal-coding spiking LLMs**.

To address this fundamental challenge, we propose a discretization-aware theoretical framework that establishes a precise correspondence between discrete TTFS-based SNNs and ANNs. Our key theoretical insight reveals that conversion errors are formally bounded by latency-dependent terms, drawing a direct connection to quantization error bounds. Motivated by this equivalence, we introduce a paradigm shift from traditional continuous-approximation conversions to discrete-equivalent coversion. Specifically, we present the Quantization-Consistent ANN-to-SNN Conversion (QC-A2S), which integrates low-bit quantization with discretization-compatible TTFS neurons. QC-A2S leverages pre-quantized LLMs to inherently align with discrete spike dynamics, effectively mitigating conversion errors while achieving latency-efficient temporal-coding spiking LLMs. Comprehensive evaluation on LLaMA models demonstrates that our approach maintains comparable accuracy with dramatically reduced inference latency (Figure 2). The key contributions are summarized as follows:

- We propose a discretization-aware theoretical framework for TTFS-based coding that identifies the fundamental discrepancy between continuous-time assumptions in prior TTFS methods and practical hardware constraints, revealing the formal equivalence between conversion errors and quantization error bounds.

- We present the QC-A2S framework, which represents a paradigm shift from traditional continuous-approximation conversions to discrete-equivalent transformation, enabling the first latency-efficient TTFS-based temporal-coding spiking LLMs.

- Extensive experiments on LLaMA models demonstrate that our framework successfully constructs temporal spiking LLMs with performance comparable to their original counterparts while achieving significant latency reduction.

## 2 RELATED WORKS

### 2.1 SPIKING LLMS

The success of LLMs has motivated the development of SNN counterparts (spiking LLMs) that maintain energy efficiency while achieving comparable capabilities. Several approaches have

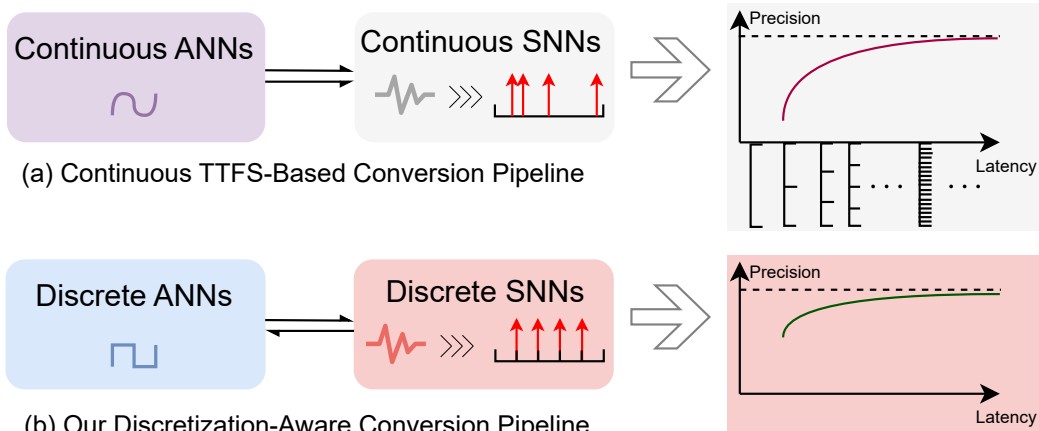

Figure 2: Overview of the QC-A2S framework. By establishing a discretization-aware equivalence between quantized ANNs and discrete TTFS SNNs, QC-A2S eliminates the latency-dependent conversion errors inherent in continuous-time pipelines and enables accurate, quantization-consistent spiking LLMs.

emerged for creating spiking variants of transformer-based models (You et al., 2024; Zhou et al., 2022; 2023). SpikeGPT replaces traditional self-attention with Spiking RWKV mechanisms (Zhu et al., 2023). SpikingBERT employs a two-stage knowledge distillation method that utilizes pre-trained BERT models as teachers to train spiking student architectures (Lv et al., 2023). Similarly, SpikingMiniLM builds upon BERT with parameter initialization and ANN-to-SNN distillation methods to achieve faster convergence during training. Recent work introduced SpikeLLM, scaling to 70 billion parameters through spike-driven quantization (Xing et al., 2024b;a). However, existing spiking LLMs rely exclusively on rate coding, where information is encoded through spike frequency. This leaves unexplored the potential of temporal-based spiking LLMs, which could achieve substantially lower energy consumption.

## 2.2 TEMPORAL-BASED A2S CONVERSIONS

While rate-based conversion methods have dominated ANN-to-SNN conversion research, temporal-based encoding approaches offer compelling advantages in terms of energy efficiency by leveraging precise spike timing rather than spike frequency. These methods include time-to-first spike (Thorpe et al., 2001), reverse coding (Zhang et al., 2019; Park et al., 2020), phase coding (Montemurro et al., 2008) and burst coding (Park et al., 2019). Among temporal coding schemes, time-to-first-spike (TTFS) coding has emerged as particularly promising, where each neuron emits at most one spike per time window with information encoded in the spike latency. Early TTFS-based conversion methods were developed by Rueckauer & Liu (2018) and further improved by Zhang et al. (2019) and Park et al. (2020), but these approaches introduced conversion errors across layers. A breakthrough came with Stanojevic et al. (2023; 2024), who demonstrated exact mapping from ReLU-based networks to SNNs using TTFS coding through a two-stage neuron activation process, achieving lossless conversion while maintaining energy benefits. Recently, Zhao et al. (2025) proposed TTFSFormer, the first TTFS-based conversion framework for Transformer architectures. However, existing TTFS-based conversion methods require extremely high latency to match continuous-time assumptions, preventing their implementation on large-scale models.

## 2.3 MODEL QUANTIZATION

Quantization has emerged as a critical technique for reducing model size and memory consumption, enabling efficient deployment of LLMs on resource-constrained devices (Shao et al., 2024), falling into two primary categories: quantization-aware training (QAT) (Liu et al., 2023) and post-training quantization (PTQ) (Xiao et al., 2023). QAT optimizes quantized weights during training using techniques like the straight-through estimator (Chen et al., 2024c; Du et al., 2024) but is computationally impractical for LLMs. PTQ has thus become the preferred approach, requiring only minimal calibration data while using dynamic activation quantization to address outlier-induced ac-

curacy degradation (Frantar et al., 2023). Recent PTQ advances address outlier-induced errors using orthogonal transformations (QuaRot (Ashkboos et al., 2024), SpinQuant (Liu et al., 2024)) or dual transformations (DuQuant (Lin et al., 2024)) to redistribute outliers across channels. However, these methods require computationally expensive per-token dynamic computation during inference. PrefixQuant (Chen et al., 2024b) offers an alternative by isolating token-wise outliers to enable efficient per-tensor static quantization, achieving comparable performance. While these quantization methods successfully achieve competitive performance with low-bit representations, energy consumption from dense matrix operations remains a fundamental barrier to edge deployment.

## 3 REVISITING TTFS-BASED ANN-TO-SNN CONVERSION

### 3.1 CONTINUOUS TTFS-BASED NEURONS

The activation process of continuous TTFS-based neurons is generally divided into two stages: the receiving phase and the firing phase (Zhao et al., 2025). At the $i$-th neuron in $l$-th layer, $i = 1, 2, ..., I$ and $l = 1, 2, ..., L$. We denote the time range of the receiving phase as $[t_{\text{recv}}^{(l)}, t_{\text{emit}}^{(l)}]$, and the emitting phase as $[t_{\text{emit}}^{(l)}, t_{\text{end}}^{(l)}]$. With the initial membrane potential $V(t_{\text{recv}}^{(l)}) = 0$, the continuous membrane potential dynamics are given by:

$$\frac{d}{dt}V(t) = \begin{cases} \frac{1}{\tau_i^{(l)}} \left( \sum_j w_{ij}^{(l)} \eta_{ij}^{(l)} \left( t - t_j^{(l-1)} \right) + C_i^{(l)} \right), & t \in [t_{\text{recv}}^{(l)}, t_{\text{emit}}^{(l)}), \\ \psi_i^{(l)} \left( t - t_{\text{emit}}^{(l)} \right), & t \in [t_{\text{emit}}^{(l)}, t_{\text{end}}^{(l)}). \end{cases} \tag{1}$$

The spike time $t_j^{(l-1)}$ is received from the previous layer of the $j$-th input, while using the time range of the receiving phase from the previous layer as the time range for the firing phase of this layer, i.e., $t_{\text{recv}}^{(l)} = t_{\text{emit}}^{(l-1)}$ and $t_{\text{emit}}^{(l)} = t_{\text{end}}^{(l-1)}$; $w_{ij}^{(l)}$ are the weights; the input transform kernel function $\eta_{ij}^{(l)}$ satisfies $\eta_{ij}^{(l)}(u) = 0$, $\forall u < 0$; $\tau_i^{(l)} > 0$ is the time constant; $C_i^{(l)}$ serves as a bias term; the output transform kernel function $\psi_i^{(l)}$ is non-negative. Once the potential exceeds the threshold $\theta_i^{(l)}$, the neuron will emit a spike and record the spike firing time $t_i^{(l)}$. The relation between the spike time $t_i^{(l)}$ and the corresponding activation value $x_i^{(l)}$ of ANNs is:

$$x_i^{(l)} \tau_i^{(l)} = t_{\text{ref}}^{(l)} - t_i^{(l)}, \tag{2}$$

where $t_{\text{ref}}^{(l)}$ is the zero reference time. Therefore, the output range $[a_i^{(l)}, b_i^{(l)}]$ can be expressed as:

$$a_i^{(l)} = \frac{1}{\tau_i^{(l)}} \left( t_{\text{ref}}^{(l)} - t_{\text{end}}^{(l)} \right), \qquad b_i^{(l)} = \frac{1}{\tau_i^{(l)}} \left( t_{\text{ref}}^{(l)} - t_{\text{emit}}^{(l)} \right). \tag{3}$$

We denote $T^{(l)} = t_{\text{end}}^{(l)} - t_{\text{emit}}^{(l)}$ as the time window, and $d_i^{(l)} = b_i^{(l)} - a_i^{(l)}$.

### 3.2 PRACTICAL LIMITATIONS OF CONTINUOUS TTFS-BASED CONVERSION

The continuous TTFS-based conversion method (Zhao et al., 2025) establishes an equivalence between TTFS-based neurons and ANN neurons by modifying the input and output transform kernel functions, thereby enabling the mapping of TTFS-based SNNs to continuous ANNs:

**Theorems 4.1 and 4.3** in Zhao et al. (2025): Let $f_{ij} : [a_i^{(l-1)}, b_i^{(l-1)}] \to \mathbb{R}$ be differentiable functions and $h : A \to \mathbb{R}$ be a differentiable monotone increasing function, and its inverse $h^{-1}$ is well-defined on $[a_i^{(l)}, b_i^{(l)}]$. If we let

$$\eta_{ij}^{(l)}(s) = \begin{cases} f_{ij}' \left( \frac{s}{\tau_i^{(l-1)}} + a_i^{(l-1)} \right), & s \geq 0, \\ 0, & s < 0, \end{cases}, \quad C_i^{(l)} = \sum_j w_{ij} \frac{f_{ij}\left( a_i^{(l-1)} \right)}{d_i^{(l-1)}},$$

$$\psi_i^{(l)}(s) = \frac{1}{\tau_i^{(l)} h' \left( h^{-1} \left( b_i^{(l)} - \frac{s}{\tau_i^{(l)}} \right) \right)}, \quad \theta_i^{(l)} = h^{-1}(b_i^{(l)}) \tag{4}$$

then the value $x_i^{(l)}$ of ANNs represented by the output spike is

$$x_i^{(l)} = f^{(l)}(W^{(l)}; x_1^{(l-1)}, ..., x_I^{(l-1)}) = \text{clip}\left(h\left(\sum_j w_{ij}^{(l)} f_{ij}\left(x_j^{(l-1)}\right)\right), a_i^{(l)}, b_i^{(l)}\right) \quad (5)$$

Although TTFS-based ANN-to-SNN conversion methods under continuous setting have been explored, their applications to LLMs remain limited in two aspects:

**Infinite Clock Precision:** For TTFS-based neurons under continuous setting , the spike time can be any real number (Stanojevic et al., 2023; Zhao et al., 2025; Stanojevic et al., 2024). At this point, the required clock precision is theoretically infinitely fine: $\Delta t_{real} \to 0$. However, electronic neuromorphic chips, which rely on discrete clock cycles, cannot provide infinitely fine clock precision (Deng et al., 2023). Consequently, TTFS coding based on continuous assumptions faces significant limitations in hardware implementations.

**Latency Overhead of Lossless Conversion:** In the continuous setting, TTFS-based lossless conversion methods establish **an equivalence between SNNs and continuous ANNs** and directly mapping the former to the latter. However, this process incurs extremely high latency (e.g., up to 4096 time steps), which propagates through the network and leads to prohibitively long inference delays.

> In continuous settings, TTFS coding requires prohibitively high latency to achieve lossless conversion, resulting in excessively long inference delays for LLMs.

## 4 DISCRETIZATION-AWARE CONVERSION

In this section, we first construct discrete TTFS-based neurons to address the challenge of infinite clock precision. Next, rather than directly mapping TTFS-based SNNs to continuous ANNs in a continuous setting, we analyze the relationship between TTFS-based SNNs and discrete ANNs. We then examine the conversion error of discrete TTFS-based SNNs. Finally, we introduce the Quantization-Consistent ANN-to-SNN conversion method.

### 4.1 DISCRETE TTFS-BASED NEURONS

To overcome the challenge posed by infinite clock precision, we constructed a hardware-friendly discrete TTFS coding neuron model. Under the discrete time-step setting, the differential form of the original membrane potential equation can be approximated as follows:

$$\frac{d}{dt}V(t) = \frac{dV(t)}{dt_{real}(t)} \cdot \frac{dt_{real}(t)}{dt} \approx \frac{V(t+1) - V(t)}{t_{real}(t+1) - t_{real}(t)} \cdot \frac{d}{dt}t_{real}(t) = V(t+1) - V(t). \quad (6)$$

Building on the above discussion, we present a discretized version of TTFS-based neurons. At the $i$-th neuron in $l$-th layer, $i = 1, 2, ..., I$ and $l = 1, 2, ..., L$. We denote the time range of the receiving phase as $\{t_{\text{recv}}^{(l)}, ..., t_{\text{emit}}^{(l)}\}$, and the emitting phase as $\{t_{\text{emit}}^{(l)}, ..., t_{\text{end}}^{(l)}\}$. With the initial membrane potential $V(t_{\text{recv}}^{(l)}) = 0$, the discrete membrane potential dynamics are given by:

$$V(t+1) - V(t) = \begin{cases} \frac{1}{\tau_i^{(l)}}\left(\sum_j w_{ij}^{(l)}\eta_{ij}^{(l)}\left(t - t_j^{(l-1)}\right) + C_i^{(l)}\right) & t \in \{t_{\text{recv}}^{(l)}, ..., t_{\text{emit}}^{(l)} - 1\}, \\ \psi_i^{(l)}\left(t - t_{\text{emit}}^{(l)}\right) & t \in \{t_{\text{emit}}^{(l)}, ..., t_{\text{end}}^{(l)} - 1\}. \end{cases} \quad (7)$$

We denote $T^{(l)} = t_{\text{end}}^{(l)} - t_{\text{emit}}^{(l)}$ as the time window, and $d_i^{(l)} = b_i^{(l)} - a_i^{(l)}$.

### 4.2 RELATIONSHIP BETWEEN DISCRETE TTFS-BASED SNNS AND ANNS

We theoretically establish the equivalence between TTFS-based SNNs and discrete ANNs. First, we determine the corresponding ANN function using the transform kernel functions and parameters of the TTFS-based neuron. For any TTFS-based neuron with fixed conversion functions and parameters, the corresponding ANN function can be identified:

**Theorem 1** *For arbitrary fixed $\eta_{ij}^{(l)}$, $\psi_i^{(l)}$, $C_i^{(l)}$ and $\theta_i^{(l)}$ in SNNs with time window $T^{(l)}$, if we define $S(t) = \sum_{v=0}^{t-t_{emit}^{(l)}-1} \psi_i^{(l)}(v)$ with $t \in \left\{ t_{emit}^{(l)}, \ldots, t_{end}^{(l)} \right\}$, then the corresponding activation value of discrete ANNs is given by:*

$$x_i^{(l)} = f^{(l)}(W^{(l)}; x_1^{(l-1)}, \ldots, x_I^{(l-1)}) \tag{8}$$

$$= \frac{1}{\tau_i^{(l)}} \left( t_{ref}^{(l)} - S^{-1} \left( \theta_i^{(l)} + \Delta_i^{(l)} - \frac{1}{\tau_i^{(l)}} \sum_{j=1}^{I} \sum_{t=t_j^{(l-1)}}^{T_{emit}^{(l)}-1} w_{ij}^{(l)} \eta_{ij}^{(l)} \left( x_j^{(l-1)} \tau_j^{(l-1)} + t - t_{ref}^{(l-1)} \right) - T^{(l)} C_i^{(l)} \right) \right),$$

*where $W^{(l)} = (w_{ij}^{(l)})_{I \times I}$ is the weight matrix; $\Delta_i^{(l)} \geq 0$ is a compensation constant, which is actually the difference between the $\theta_i^{(l)}$ and the membrane potential at the spike time.*

Next, we determine the corresponding transform kernel functions and parameters in the TTFS-based neuron using the ANN function. For any given fixed ANN function, the TTFS-based neuron with the corresponding transform kernel functions and parameters can be identified:

**Theorem 2** *Let $f_{ij}$ be a function with input set of discrete points between $a_i^{(l-1)}$ and $b_i^{(l-1)}$, and $h$ be a monotone increasing function with output set of discrete points between $a_i^{(l)}$ and $b_i^{(l)}$. We denote $u = t - t_j^{(l-1)}$ with $t \in \left\{ t_{recv}^{(l)}, \ldots, t_{emit}^{(l)} \right\}$, and $v = t - t_{emit}^{(l)}$ with $t \in \left\{ t_{emit}^{(l)}, \ldots, t_{end}^{(l)} \right\}$. To represent the corresponding activation value of discrete ANNs:*

$$x_i^{(l)} = f^{(l)}(W^{(l)}; x_1^{(l-1)}, \ldots, x_I^{(l-1)}) = \text{clip} \left( h \left( \sum_j w_{ij}^{(l)} f_{ij} \left( x_j^{(l-1)} \right) \right), a_i^{(l)}, b_i^{(l)} \right). \tag{9}$$

*we need to configure the SNN as follows:*

$$\eta_{ij}^{(l)}(u) = \begin{cases} \tau_i^{(l-1)} \left( f_{ij} \left( \frac{u+1}{\tau_i^{(l-1)}} + a_i^{(l-1)} \right) - f_{ij} \left( \frac{u}{\tau_i^{(l-1)}} + a_i^{(l-1)} \right) \right) & u \geq 0, \\ 0 & u < 0. \end{cases}$$

$$\psi_i^{(l)}(v) = h^{-1} \left( b_i^{(l)} - \frac{v}{\tau_i^{(l)}} \right) - h^{-1} \left( b_i^{(l)} - \frac{v+1}{\tau_i^{(l)}} \right), \quad C_i^{(l)} = \frac{\sum_j w_{ij}^{(l)} f_{ij}\left( a_i^{(l-1)} \right)}{d_i^{(l-1)}}, \quad \theta_i^{(l)} = h^{-1}(b_i^{(l)}) + \Delta_i^{(l)}.$$

Furthermore, we demonstrate the equivalence between the discrete TTFS-based neuron and the quantization function:

**Corollary 1** *We define the processes of quantization and dequantization as follows:*

$$\hat{\mathbf{X}}_i^{(l)} = \lambda_i^{(l)} \cdot \text{clip}(\lfloor \frac{\mathbf{X}_i^{(l)}}{\lambda_i^{(l)}} \rfloor + z^{(l)}, 0, N) - \lambda_i^{(l)} \cdot z^{(l)}, \tag{10}$$

*where $\lambda_i^{(l)} = \frac{\max(\mathbf{X}_i^{(l)}) - \min(\mathbf{X}_i^{(l)})}{N}$ and $z^{(l)} = -\lfloor \frac{\min(\mathbf{X}_i^{(l)})}{\lambda_i^{(l)}} \rfloor$ are scale and zero point values, respectively; $\lfloor \cdot \rfloor$ denotes the floor operation; $N = 2^n - 1$ denotes the quantization level and $n$ denotes the quantization bits; $\hat{\mathbf{X}}_i^{(l)}$ and $\mathbf{X}_i^{(l)}$ are the dequantized and original tensor, respectively.*

*For a TTFS-based SNN defined in (7), $\mathcal{H}$ is the Heaviside step function, if we set the $\eta_{ij}^{(l)}$, $\psi_i^{(l)}$, $C_i^{(l)}$ and $\theta_i^{(l)}$ as follow:*

$$\eta_{ij}^{(l)}(u) = \mathcal{H} \left( \frac{u}{\tau_i^{(l-1)}} + a_i^{(l-1)} \right), \quad \psi_i^{(l)}(v) = \frac{1}{\tau_i^{(l)}}, \quad C_i^{(l)} = \sum_j \frac{a_i^{(l-1)}}{d_i^{(l-1)}} w_{ij}, \quad \theta_i^{(l)} = b_i^{(l)} \tag{11}$$

*and we let $t_{emit}^{(l)} = 0$, $t_{end}^{(l)} = N$, $\tau_i^{(l)} = \frac{1}{\lambda_i^{(l)}}$, $t_{end}^{(l)} - t_{ref}^{(l)} = z^{(l)}$, and $\mathbf{X}_i^{(l)} = \sum_{j=1}^{I} w_{ij}^{(l)} x_j^{(l-1)}$. The output of spiking neural neuron and quantization function are equivalent, i.e, $x_i^{(l)} = \hat{\mathbf{X}}_i^{(l)}$.*

## 4.3 ERROR ANALYSIS FOR DISCRETE TTFS-BASED SNNS

In the continuous setting, although TTFS-based SNNs enable lossless conversion to ANNs, they require infinitely fine clock precision for hardware implementation and introduce significantly long inference delay in the network. We analyze the conversion error of discrete TTFS-based SNNs.

**Theorem 3** *The error analysis of TTFS-based SNNs: Let $T^{(l)}$ denotes the time window with the corresponding clock time constant $\Omega$, the derivatives of the function $h$ and its inverse are bounded by $G_1$ and $G_2$, $I$ denotes the number of neurons in each layer of the network, and $L$ denotes the number of layers, $T = \min\left\{T^{(l)}\right\}_{l=1}^{L}$, and $\tau = \max\left\{\left\{\tau_i^{(l)}\right\}_{i=1}^{I}\right\}_{l=1}^{L}$, $\alpha_i^{(l)}$ is the corresponding output of ANNs and $\rho = \max_{\{i,l\}}\left\{\left|\alpha_i^{(l)} - \frac{a_i^{(l)}+b_i^{(l)}}{2}\right|\right\}$. The conversion error of the TTFS-based SNNs in can be bounded as:*

$$\mathcal{E} \leq LI \cdot \max\left(\rho - \frac{T}{2\tau}, 0\right) + \frac{LIG_1G_2\Omega}{T} \tag{12}$$

**Remark 1** *In Theorem 3: The first term captures the clipping error in the TTFS-based SNNs, which can be eliminated by increasing the time window $T$. As $T$ increases, the output range of TTFS-based SNNs expands. When this range encompasses the output of ANNs, the clipping error is eliminated; The second term reflects the quantization error, which can only be alleviated by increasing $T$. As $T$ increases, the output range of TTFS-based SNNs becomes finer, facilitating better alignment between the output of ANNs and the discrete points of the SNNs' output, thereby reducing quantization error. Thus, achieving high accuracy TTFS-based SNNs necessitates sufficiently long time windows.*

## 4.4 QUANTIZATION-CONSISTENT ANN-TO-SNN CONVERSION

Our goal is to develop high-accuracy, low-latency temporal-coding spiking LLMs. Achieving high accuracy in temporal-coding spiking LLMs typically requires extending the time window, which in turn increases latency. This latency propagates through the network, leading to excessive inference delays. To address this challenge, we propose the Quantization-Consistent ANN-to-SNN (QC-A2S) conversion method, which leverages the equivalence between TTFS-based SNNs and discrete ANNs. Our approach combines low-bit quantization with discretization-compatible TTFS neurons, enabling low-latency temporal-coding spiking LLMs. Specifically, we first apply established techniques, such as post-training quantization, to minimize clipping and quantization errors, resulting in a low-bit, high-accuracy baseline model. We then map the quantized LLM to an equivalent spiking LLM, achieving a low-latency, high-accuracy temporal-coding spiking LLM.

## 5 EXPERIMENT

In this section, we conduct experiments to validate the effectiveness of our proposed method and compare its performance, computational count, and energy consumption with those of different approaches. Additionally, we conduct ablation studies on various latency.

### 5.1 IMPLEMENT DETAILS

**Datasets and Underlying Models**  In the experiments, two types of benchmarks are used. For accuracy-oriented evaluation, five representative reasoning datasets are adopted, namely PIQA(Bisk et al., 2020), HellaSwag (Zellers et al., 2019), WinoGrande (Sakaguchi et al., 2021), ARC-Easy, and ARC-Challenge (Clark et al., 2018). **PIQA** targets physical commonsense reasoning in everyday scenarios, **ARC-Easy** and **ARC-Challenge** consist of science exam questions with varying difficulty levels, **HellaSwag** evaluates contextual understanding through plausible continuation tasks, and **WinoGrande** focuses on large-scale pronoun resolution for commonsense reasoning. For perplexity-oriented evaluation, we additionally use five widely adopted language modeling datasets, including C4 (Raffel et al., 2020), The Pile (Gao et al., 2021), Penn Treebank (PTB) (Marcus et al., 1993), WikiText-2 (Merity et al., 2017), and RedPajama (Together Computer, 2023). The datasets were preprocessed following standard practices, and data augmentation techniques were applied

where appropriate. In our study, all methods are applied to the LLaMA family of LLMs as the common backbone. We consider a range of representative models, including LLaMA-2-7B, LLaMA-2-13B, LLaMA-3-8B, and LLaMA-2-70B.

**Baselines**  We compare our approach against several representative baselines that adapt large language models through either quantization or ANN-to-SNN conversion:

- **PrefixQuant** (Chen et al., 2025b) is a weight–activation quantization method that addresses token-wise outliers in the KV cache and employs lightweight blockwise training, achieving strong performance across different precision levels.

- **SpikeLLM** (Xing et al., 2024a) presents the first spiking LLMs by incorporating bio-inspired spiking mechanisms with generalized integrate-and-fire neurons, yielding improvements in perplexity and reasoning accuracy compared to quantized LLMs.

- **TTFSFormer** (Zhao et al., 2025) applies time-to-first-spike coding to Transformers, extending TTFS neurons to handle nonlinear layers and achieving competitive accuracy with significantly reduced energy consumption.

**Experiment Configurations**  All experiments were conducted on a server equipped with NVIDIA A100 GPUs (80 GB of memory), Intel Xeon CPUs, and 512 GB of RAM. The models were implemented in PyTorch 2.6 with CUDA 12.4 support. For fair comparison, all baseline methods were re-implemented or run using their officially released code under the same environment and hyperparameter settings whenever possible. In addition to the hardware information mentioned in the main text, we provide further details about the reproduction of baselines here. We adopt 8 bits for weight, 6 bits for activation quantization, *i.e.* W8A6, for SpikeLLM(Xing et al., 2024a) and PrefixQuant(Chen et al., 2025b), and use 8192 time precision for TTFSFormer (Zhao et al., 2025).

## 5.2 MAIN RESULTS

Tables 1 and 2 report the accuracy and PPL metrics of all methods on the LLaMA-2-7B, LLaMA-2-13B, LLaMA-3-8B, LLaMA-2-70B and LLaMA-3-70B models. The results indicate that: (i) temporally encoded spiking LLMs achieve performance comparable to quantized LLMs across all LLaMA models, providing further empirical evidence for the equivalence between TTFS-based SNNs and quantized ANNs; (ii) our method substantially outperforms TTFSFormer under low-latency settings, while TTFSFormer continues to exhibit unsatisfactory performance even at higher latencies, underscoring the excessive latency demands of continuous TTFS-based SNNs; and (iii) our model surpasses the state-of-the-art spiking LLM (SpikeLLM), further validating the effectiveness of the proposed approach.

## 5.3 COMPARISON OF ACCURACY UNDER DIFFERENT LATENCY CONFIGURATIONS

In Table 3, we conduct a detailed study of how latency influences the performance of temporally coded spiking LLMs using the LLaMA-2-7B and LLaMA-2-13B model. The results reveal a clear trend: increasing latency consistently improves accuracy across all evaluated benchmarks. This indicates that longer time windows allow TTFS-based SNNs to better approximate the activations of ANNs, thereby reducing discretization-induced errors and enhancing representational fidelity. Moreover, deeper/larger models do not necessarily yield monotonically higher accuracy. While additional layers increase model capacity, Theorem 3 shows that the approximation error bound grows with the number of layers $L$. As $L$ becomes large, the accumulated discretization error can counteract the gains from the increased capacity, which explains why scaling from 7B to 13B to 70B does not produce consistent accuracy improvements in our experiments (Table 1 and 2). Such evidences provide strong empirical support for our theoretical analysis in Theorem 3 ($\mathcal{E} \leq LI \cdot \max\left(\rho - \frac{T}{2\tau}, 0\right) + \frac{LIG_1 G_2 \Omega}{T}$), which establishes that achieving high accuracy in TTFS-based SNNs is inherently dependent on sufficiently long latency ($T$).

Table 1: We report *accuracy* for WinoGrande and *acc_norm* for HellaSwag, ArcC, ArcE, and PIQA on LLaMA-2-7B, LLaMA-2-13B, LLaMA-3-8B, LLaMA-2-70B and LLaMA-3-70B models

| Model | Method | Precision | WinoGrande | HellaSwag | ArcC | ArcE | PiQA | Avg. |
|-------|--------|-----------|------------|-----------|------|------|------|------|
| 2-7B | Baseline | FP16 | 69.22 | 76.00 | 46.33 | 74.62 | 79.11 | 69.06 |
| | TTFSFormer | T=32 | 51.07 | 26.32 | 28.84 | 25.84 | 48.91 | 36.20 |
| | TTFSFormer | T=64 | 48.86 | 26.40 | 28.67 | 26.09 | 49.24 | 35.85 |
| | TTFSFormer | T=8192 | 50.04 | 25.49 | 26.88 | 26.81 | 50.82 | 36.01 |
| | SpikeLLM | W8A6 | 65.51 | 73.61 | 42.49 | 70.16 | 75.41 | 65.44 |
| | PrefixQ | W4A4 | 66.77 | 73.62 | 42.83 | 70.88 | 76.93 | 66.21 |
| | PrefixQ | W5A5 | 69.06 | 75.53 | 43.94 | 73.06 | 77.86 | 67.89 |
| | PrefixQ | W6A6 | 70.48 | 76.22 | 45.48 | 73.86 | 78.35 | 68.88 |
| | Ours | T=16 | 66.77 | 73.08 | 41.21 | 70.45 | 77.64 | 65.83 |
| | Ours | T=32 | 69.38 | 75.49 | 44.28 | 72.85 | 78.02 | 68.00 |
| | Ours | T=64 | 69.38 | 76.23 | 45.99 | 73.57 | 78.13 | 68.66 |
| 3-8B | Baseline | FP16 | 72.69 | 79.19 | 53.41 | 77.69 | 80.79 | 72.75 |
| | TTFSFormer | T=32 | 49.88 | 26.40 | 26.54 | 24.71 | 51.58 | 35.82 |
| | TTFSFormer | T=64 | 50.59 | 26.19 | 26.11 | 24.49 | 50.16 | 35.51 |
| | TTFSFormer | T=8192 | 52.41 | 26.86 | 25.77 | 24.75 | 51.09 | 36.18 |
| | SpikeLLM | W8A6 | 58.25 | 59.28 | 32.34 | 53.37 | 68.66 | 54.38 |
| | PrefixQ | W4A4 | 71.03 | 74.51 | 48.72 | 75.88 | 77.80 | 69.59 |
| | PrefixQ | W5A5 | 71.74 | 77.59 | 53.41 | 78.45 | 79.11 | 72.06 |
| | PrefixQ | W6A6 | 72.77 | 78.52 | 53.07 | 78.58 | 79.22 | 72.43 |
| | Ours | T=16 | 70.09 | 74.36 | 48.29 | 75.97 | 77.86 | 69.31 |
| | Ours | T=32 | 70.01 | 77.64 | 53.50 | 78.07 | 79.71 | 71.79 |
| | Ours | T=64 | 72.06 | 78.26 | 54.95 | 77.90 | 79.43 | 72.52 |
| 2-13B | Baseline | FP16 | 72.38 | 79.38 | 49.06 | 77.53 | 80.52 | 71.77 |
| | TTFSFormer | T=32 | 48.15 | 26.40 | 28.24 | 25.00 | 49.02 | 35.36 |
| | TTFSFormer | T=64 | 48.54 | 26.15 | 30.03 | 26.30 | 51.03 | 36.41 |
| | TTFSFormer | T=8192 | 48.70 | 26.29 | 26.11 | 25.72 | 51.25 | 35.61 |
| | SpikeLLM | W8A6 | 68.03 | 76.76 | 44.88 | 73.32 | 77.48 | 68.09 |
| | PrefixQ | W4A4 | 69.69 | 75.76 | 47.53 | 73.78 | 78.62 | 69.08 |
| | PrefixQ | W5A5 | 72.38 | 78.30 | 49.91 | 76.26 | 79.76 | 71.32 |
| | PrefixQ | W6A6 | 72.53 | 79.07 | 49.23 | 75.76 | 79.71 | 71.26 |
| | Ours | T=16 | 70.32 | 75.62 | 46.42 | 73.95 | 77.97 | 68.86 |
| | Ours | T=32 | 71.35 | 78.24 | 49.23 | 75.00 | 79.87 | 70.74 |
| | Ours | T=64 | 72.53 | 79.07 | 49.23 | 75.76 | 79.71 | 71.26 |
| 2-70B | Baseline | FP16 | 77.98 | 83.82 | 57.42 | 80.98 | 82.70 | 76.58 |
| | TTFSFormer | T=32 | 48.15 | 26.40 | 28.24 | 25.00 | 49.02 | 35.36 |
| | SpikeLLM | W8A6 | 75.06 | 81.42 | 52.82 | 75.29 | 80.58 | 73.03 |
| | PrefixQ | W4A4 | 75.45 | 74.32 | 46.08 | 72.47 | 77.53 | 68.40 |
| | Ours | T=16 | 73.95 | 79.22 | 51.96 | 76.94 | 80.09 | 72.43 |
| 3-70B | Baseline | FP16 | 80.35 | 84.88 | 64.33 | 85.86 | 84.55 | 79.99 |
| | TTFSFormer | T=32 | 49.01 | 26.32 | 26.88 | 25.25 | 50.05 | 35.50 |
| | SpikeLLM | W8A6 | 52.09 | 28.40 | 26.45 | 31.36 | 53.59 | 38.38 |
| | PrefixQ | W4A4 | 77.98 | 82.84 | 59.98 | 81.73 | 83.35 | 77.18 |
| | Ours | T=16 | 77.27 | 82.67 | 59.04 | 82.15 | 83.13 | 76.85 |

Table 2: We report *Perplexity* for C4, Pile, PTB, WikiText2, and RedPajama on LLaMA-2-7B, LLaMA-2-13B, LLaMA-3-8B, LLaMA-2-70B and LLaMA-3-70B models

| Model | Method | Precision | C4 | Pile | PTB | WikiText2 | RedPajama | Avg. |
|-------|--------|-----------|------|------|------|-----------|-----------|------|
| 2-7B | Baseline | FP16 | 6.97 | 4.63 | 37.91 | 5.47 | 5.61 | 12.12 |
| | TTFSFormer | T=32 | >100 | >100 | >100 | >100 | >100 | >100 |
| | TTFSFormer | T=64 | >100 | >100 | >100 | >100 | >100 | >100 |
| | TTFSFormer | T=8192 | >100 | >100 | >100 | >100 | >100 | >100 |
| | SpikeLLM | W8A6 | 7.89 | 5.14 | 57.27 | 6.43 | 6.21 | 16.59 |
| | PrefixQ | W4A4 | 7.72 | 5.00 | 33.01 | 6.12 | 6.28 | 11.63 |
| | PrefixQ | W5A5 | 7.20 | 4.74 | 32.16 | 5.67 | 5.82 | 11.12 |
| | PrefixQ | W6A6 | 7.06 | 4.67 | 67.30 | 5.54 | 5.70 | 18.06 |
| | Ours | T=16 | 7.73 | 5.00 | 33.00 | 6.12 | 6.29 | 11.63 |
| | Ours | T=32 | 7.20 | 4.74 | 32.00 | 5.67 | 5.82 | 11.09 |
| | Ours | T=64 | 7.07 | 4.67 | 67.37 | 5.54 | 5.70 | 18.07 |
| 3-8B | Baseline | FP16 | 8.88 | 5.52 | 11.18 | 6.14 | 7.44 | 7.83 |
| | TTFSFormer | T=32 | >100 | >100 | >100 | >100 | >100 | >100 |
| | TTFSFormer | T=64 | >100 | >100 | >100 | >100 | >100 | >100 |
| | TTFSFormer | T=8192 | >100 | >100 | >100 | >100 | >100 | >100 |
| | SpikeLLM | W8A6 | >100 | >100 | >100 | >100 | >100 | >100 |
| | PrefixQ | W4A4 | 11.22 | 6.62 | 13.38 | 7.82 | 9.69 | 9.74 |
| | PrefixQ | W5A5 | 9.75 | 5.94 | 11.97 | 6.79 | 8.30 | 8.55 |
| | PrefixQ | W6A6 | 9.29 | 5.74 | 11.57 | 6.47 | 7.85 | 8.18 |
| | Ours | T=16 | 11.23 | 6.61 | 13.37 | 7.82 | 9.69 | 9.75 |
| | Ours | T=32 | 9.75 | 5.94 | 11.97 | 6.79 | 8.29 | 8.55 |
| | Ours | T=64 | 9.28 | 5.74 | 11.57 | 6.47 | 7.86 | 8.18 |
| 2-13B | Baseline | FP16 | 6.47 | 4.34 | 50.94 | 4.88 | 5.19 | 14.36 |
| | TTFSFormer | T=32 | >100 | >100 | >100 | >100 | >100 | >100 |
| | TTFSFormer | T=64 | >100 | >100 | >100 | >100 | >100 | >100 |
| | TTFSFormer | T=8192 | >100 | >100 | >100 | >100 | >100 | >100 |
| | SpikeLLM | W8A6 | 7.16 | 4.74 | 62.07 | 5.81 | 5.53 | 17.06 |
| | PrefixQ | W4A4 | 8.46 | 5.33 | 724.00 | 6.38 | 7.05 | 150.24 |
| | PrefixQ | W5A5 | 10.89 | 6.99 | 101.41 | 7.28 | 8.66 | 27.05 |
| | PrefixQ | W6A6 | 6.62 | 4.44 | 36.49 | 5.03 | 5.32 | 11.58 |
| | Ours | T=16 | 8.36 | 5.29 | 734.17 | 6.36 | 7.02 | 152.24 |
| | Ours | T=32 | 11.09 | 7.13 | 101.73 | 7.39 | 8.82 | 27.23 |
| | Ours | T=64 | 6.63 | 4.45 | 36.61 | 5.04 | 5.33 | 11.61 |
| 2-70B | Baseline | FP16 | 5.52 | 3.74 | 24.25 | 3.32 | 4.31 | 8.23 |
| | TTFSFormer | T=32 | >100 | >100 | >100 | >100 | >100 | >100 |
| | SpikeLLM | W8A6 | 6.28 | 4.18 | 24.15 | 4.97 | 4.11 | 8.74 |
| | PrefixQ | W4A4 | 7.31 | 4.76 | 31.31 | 5.02 | 5.85 | 10.85 |
| | Ours | T=16 | 7.31 | 4.76 | 31.34 | 5.03 | 5.85 | 10.86 |
| 3-70B | Baseline | FP16 | 6.73 | 4.38 | 8.52 | 2.85 | 5.32 | 5.56 |
| | TTFSFormer | >100 | >100 | >100 | >100 | >100 | >100 | >100 |
| | SpikeLLM | W8A6 | >100 | >100 | >100 | >100 | >100 | >100 |
| | PrefixQ | W4A4 | 8.18 | 5.16 | 9.71 | 4.66 | 6.92 | 6.92 |
| | Ours | T=16 | 8.16 | 5.15 | 9.65 | 4.64 | 6.90 | 6.90 |

Table 3: We report *accuracy* for WinoGrande and *acc_norm* for HellaSwag, ArcC, ArcE, and PIQA on LLaMA-2-7B, LLaMA-2-13B for ablation analysis

| Model | Latency | WinoGrande | HellaSwag | ArcC | ArcE | PIQA | Avg. |
|-------|---------|------------|-----------|------|------|------|------|
|       | T=16    | 66.77      | 73.08     | 41.21 | 70.45 | 77.64 | 65.83 |
|       | T=32    | 69.38      | 75.49     | 44.28 | 72.85 | 78.02 | 68.00 |
| 2-7B  | T=64    | 69.38      | 76.23     | 45.99 | 73.57 | 78.13 | 68.66 |
|       | T=256   | 70.56      | 76.69     | 76.69 | 73.57 | 78.40 | 69.13 |
|       | T=1024  | 70.32      | 76.62     | 46.16 | 73.78 | 78.18 | 69.01 |
|       | T=16    | 70.32      | 75.62     | 46.42 | 73.95 | 77.97 | 68.86 |
|       | T=32    | 71.35      | 78.24     | 49.23 | 75.00 | 79.87 | 70.74 |
| 2-13B | T=64    | 72.53      | 79.07     | 49.23 | 75.76 | 79.71 | 71.26 |
|       | T=256   | 72.85      | 79.71     | 49.32 | 76.64 | 80.36 | 71.78 |
|       | T=1024  | 72.38      | 79.75     | 48.72 | 76.43 | 80.52 | 71.56 |

## 6 CONCLUSION

LLMs have achieved remarkable success, but they also introduce severe energy bottlenecks that hinder their sustainable deployment. SNNs provide a promising pathway toward energy-efficient spiking LLMs through ANN-to-SNN conversion. Among various spike-coding schemes, TTFS coding is particularly appealing, as it conveys information with a single spike, thereby further reducing energy consumption. Existing TTFS-based A2S conversions depend on continuous-time assumptions and require prohibitively large latencies to approximate the continuous values of ANNs. This reliance results in unacceptable inference delays in deep models, particularly LLMs, creating significant obstacles to the development of practical temporal-coding spiking LLMs.

To overcome this challenge, we propose a discretization-aware theoretical framework that establishes a precise correspondence between discrete TTFS-based neurons and ANNs. Our key insight shows that conversion errors are constrained by latency-dependent terms. Building on this, we introduce the QC-A2S conversion method, which combines low-bit quantization with discretization-compatible TTFS neurons, enabling low-latency temporal-coding spiking LLMs.

## ETHICS STATEMENT

All participants in this work, as well as the paper submission, adhere to the ICLR Code of Ethics ( https://iclr.cc/public/CodeOfEthics).

## REPRODUCIBILITY STATEMENT

We affirm that the results of this work are fully reproducible. Appendix D provides the theoretical proofs. Appendix B.1 details the experimental implementations, and the source code will be publicly released after publication of the paper.

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

## A  NOTIONS

Table 4: Symbol Definitions

| Symbol | Definition | Symbol | Definition |
|---|---|---|---|
| $l$ | Layer index | $\mathbf{X}$ | Inputs of QANN |
| $i, j$ | Neuron index | $\hat{\mathbf{X}}$ | Output of QANN |
| $W$ | Weight matrix | $a_i^{(l)}$ | Output lower bound of TTFS-based neuron |
| $t_{\text{recv}}$ | Receiving time step | $b_i^{(l)}$ | Output upper bound of TTFS-based neuron |
| $t_{\text{emit}}$ | Emitting time step | $I$ | The number of neurons in each layer |
| $t_{\text{end}}$ | End time-step | $\theta$ | Threshold |
| $\mathcal{H}$ | Heaviside function | $C$ | Bias term in TTFS-based neuron |
| $t$ | Time step index | $\tau$ | Time constant in TTFS-based neuron |
| $\eta$ | Input transform kernel | $N$ | Quantization level |
| $\psi$ | Output transform kernel | $n$ | Quantization bits |

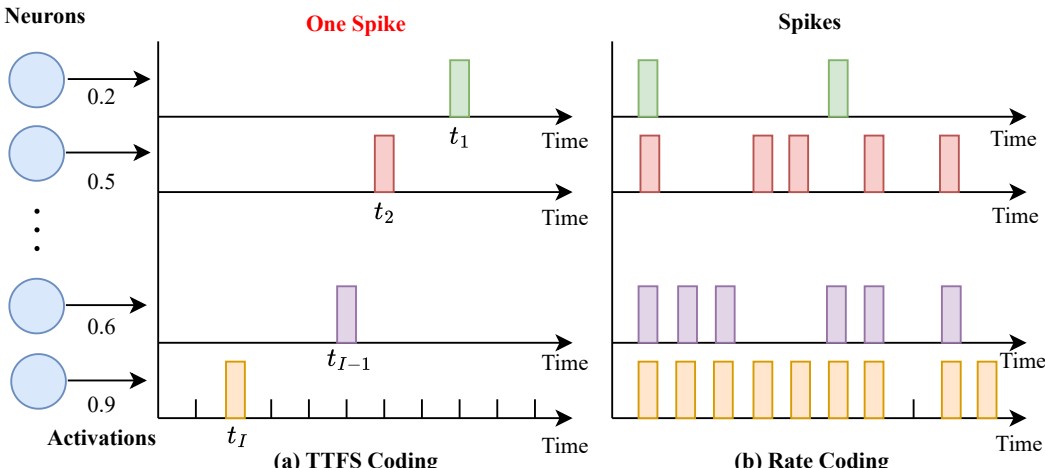

Figure 3: TTFS Coding vs. Rate Coding.

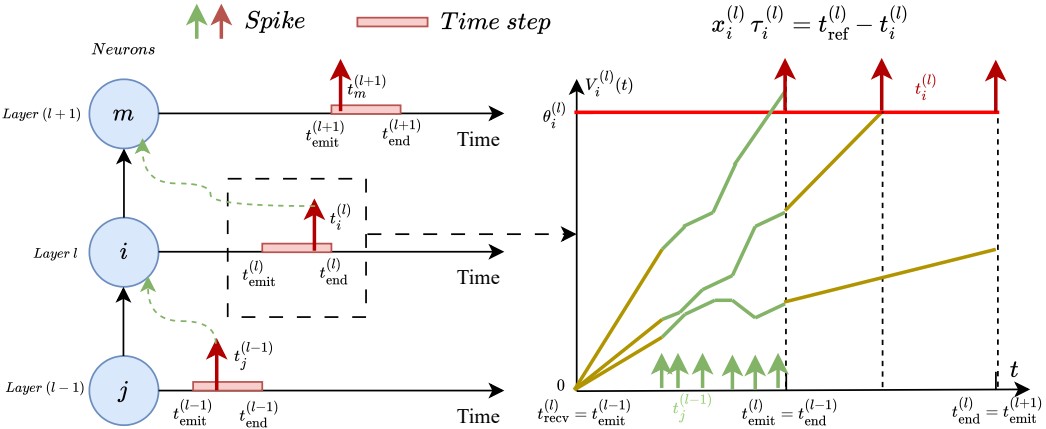

Figure 4: The process of TTFS-based spiking neural neurons.

## B  EXPERIMENTAL SUPPLEMENTARY

### B.1  EXPERIMENT CONFIGURATIONS

In addition to the hardware information mentioned in the main text, we provide further details about the reproduction of baselines here. We adopt 8 bits for weight, 6 bits for activation quantization, *i.e.* W8A6, for SpikeLLM(Xing et al., 2024a) and PrefixQuant(Chen et al., 2025b), and use 8192 time precision for TTFSFormer (Zhao et al., 2025).

## C  CONVERSION ERROR

In this section, we provide a detailed analysis of the conversion error between the ANN and the converted TTFS-based SNN across layers. We assume that both the ANN and SNN receive the same input from layer $l-1$, i.e., $\alpha^{(l-1)} = x^{(l-1)}$, and then analyze the error in layer $l$.

**ANN neurons.** For ANNs, the output $\alpha^l$ of neurons in layer $l$ is realized by a linear weighting $W^{(l)}$ and nonlinear mappings $f(\cdot)$:

$$\alpha^{(l)} = f\left(W^{(l)}\alpha^{(l-1)}\right), \qquad (13)$$

**SNN neurons.** For TTFS-based SNNs, we consider the relation between the spike time $t^{(l)}$ of SNN and the corresponding activation value $x^{(l)}$ of ANN:

$$x^{(l)} = \frac{1}{\tau^{(l)}}\left(t_{\text{ref}}^{(l)} - t^{(l)}\right). \qquad (14)$$

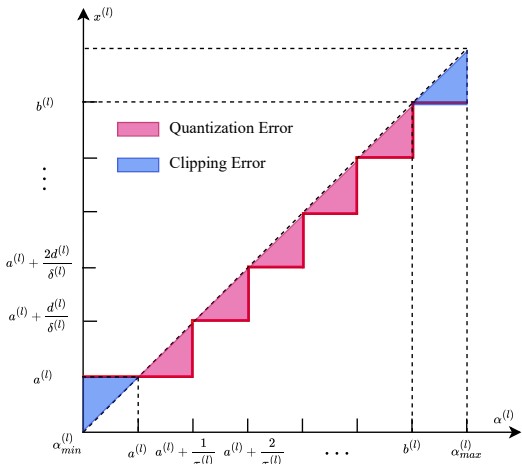

Figure 5: Clipping error and quantization error.

From Eqs.(13) and (14), along with the conditions $V\left(t_{\text{emit}}^{(l)}\right) = W^{(l)} \cdot f\left(x^{(l-1)}\right)$ and $V(t^{(l)} - 1) < \theta^{(l)} \leq V(t^{(l)})$, $t^{(l)} \in \{t_{\text{emit}}^{(l)}, \ldots, t_{\text{end}}^{(l)}\}$, it follows that a transformation between the temporal domain (relate to $t^{(l)}$) and the numerical domain (relate to $x^{(l)}$) enables the activation value $a^{(l)}$ of analog neurons in the ANN to be mapped onto $x^{(l)}$ in the TTFS-based SNN. Because the output ranges and types of SNNs and ANNs differ, conversion errors are generally unavoidable. During the ANN-to-SNN conversion, two primary sources of error, clipping error $\mathcal{E}_c^{(l)}$ and quantization error $\mathcal{E}_q^{(l)}$, both of which contribute to the performance gap between ANNs and SNNs.

For layer $l$, the total error decomposes as:

$$\mathcal{E}^{(l)} = \mathcal{E}_c^{(l)} + \mathcal{E}_q^{(l)} \qquad (15)$$

**Clipping error.** Clipping error denotes the error caused by different value ranges of ANNs and SNNs. For an temporal coding spiking neural neuron, when the time steps $T^{(l)}$ are fixed, the output of SNN: $x^{(l)}$ is in the range of $\left[a^{(l)}, b^{(l)}\right]$, where $a^{(l)} = \frac{t_{\text{ref}}^{(l)} - t_{\text{end}}^{(l)}}{\tau^{(l)}}$ and $b^{(l)} = \frac{t_{\text{ref}}^{(l)} - t_{\text{emit}}^{(l)}}{\tau^{(l)}}$. We define the $\alpha_{max}$ as the maximum value in $\alpha^{(l)}$, $\alpha_{min}$ is the minimum value in in $\alpha^{(l)}$. Then the output $\alpha \in \left[\alpha_{min}, a^{(l)}\right]$ of ANNs will be mapped to the same value $a^{(l)}$, the output $\alpha \in \left[b^{(l)}, \alpha_{max}\right]$ of ANNs will be mapped to the same value $b^{(l)}$, which will cause conversion error named clipping error $\mathcal{E}_c$.

**Quantization error.** The output spike time $t^{(l)}$ is discrete, so the final output $x^{(l)} = \frac{1}{\tau^{(l)}}\left(t_{\text{ref}}^{(l)} - t^{(l)}\right)$ is also discrete, while the output activation value $\alpha$ of the ANNs is continuous. Therefore, when mapping $\alpha^{(l)}$ to $x^{(l)}$, there will be unavoidable error related to temporal resolution, named quantization error $\mathcal{E}_q$. For example, when the output of ANNs satisfies $\alpha \in \left[\frac{t^{(l)}d^{(l)}}{T^{(l)}}, \frac{(t^{(l)}+1)d^{(l)}}{T^{(l)}}\right), t^{(l)} = t_{\text{ref}}^{(l)} - t_{\text{end}}^{(l)}, t_{\text{ref}}^{(l)} - t_{\text{end}}^{(l)} + 1, ..., t_{\text{ref}}^{(l)} - t_{\text{emit}}^{(l)} - 1$, the corresponding mapped value of SNN will be $\frac{t^{(l)}d^{(l)}}{T^{(l)}}$.

**Lemma 1** *Analysis for clipping error*. *For a target ANN's output $\alpha_i^{(l)}$, the clipping error between the output of ANN and SNN is:*

$$\mathcal{E}_c^{(l)} = \begin{cases} \left\|\alpha_i^{(l)} - b_i^{(l)}\right\| & \text{if } \alpha_i^{(l)} > b_i^{(l)} \\ 0 & \text{if } \alpha_i^{(l)} \in [a_i^{(l)}, b_i^{(l)}] \\ \left\|a_i^{(l)} - \alpha_i^{(l)}\right\| & \text{if } \alpha_i^{(l)} < a_i^{(l)}, \end{cases} \qquad (16)$$

where $a_i^{(l)} = \frac{t_{ref}^{(l)} - t_{end}^{(l)}}{\tau_i^{(l)}}$ and $b_i^{(l)} = \frac{t_{ref}^{(l)} - t_{emit}^{(l)}}{\tau_i^{(l)}}$.

**Lemma 2** *Upper bound for quantization error: In the theoretical analysis under the continuous setting of Theorem 4.1 and Theorem 4.3 in Zhao et al. (2025), we denote the output under continuous coding as $y_i^{(l)} \in [a_i^{(l)}, b_i^{(l)}]$ corresponds to the ANN with continuous outputs, while in the practically deployable discrete coding scenario, the SNN output is denoted as $x_i^{(l)} \in [a_i^{(l)}, b_i^{(l)}]$ corresponds to the ANN with discretized outputs. That is, there exists quantization error $\mathcal{E}_q^{(l)}$ between TTFS coding in the continuous setting and its practical deployment. Let $T^{(l)}$ be the time window and $\Omega$ be the corresponding clock time, and the derivatives of the function $h$ and its inverse are bounded by $G_1$ and $G_2$. Then, the quantization error can be bounded as:*

$$\mathcal{E}_q^{(l)} = \left\| x_i^{(l)} - y_i^{(l)} \right\| \leq \frac{G_1 G_2 \Omega}{T^{(l)}}. \tag{17}$$

# D PROOFS

**proof 1** *Proof of Theorem 1:*
*For arbitrary fixed $\eta_{ij}^{(l)}$, $\psi_i^{(l)}$, $C_i^{(l)}$ and $\theta_i^{(l)}$ in SNNs with time window $T^{(l)}$, in the receiving phase:*

$$V\left(t_{emit}^{(l)}\right)$$

$$= \frac{1}{\tau_i^{(l)}} \sum_{t=T_{recv}^{(l)}}^{T_{emit}^{(l)}-1} \left( \sum_{j=1}^{I} w_{ij}^{(l)} \eta_{ij}^{(l)} \left( t - t_j^{(l-1)} \right) - C_i^{(l)} \right)$$

$$= \frac{1}{\tau_i^{(l)}} \sum_{j=1}^{I} \sum_{t=t_j^{(l-1)}}^{T_{emit}^{(l)}-1} w_{ij}^{(l)} \eta_{ij}^{(l)} \left( x_j^{(l-1)} \tau_j^{(l-1)} + t - t_{ref}^{(l-1)} \right) + T^{(l)} C_i^{(l)}. \tag{18}$$

*In the emitting phase, let $\Delta_i^{(l)} \geq 0$ is a compensation constant, which is actually the difference between the $\theta_i^{(l)}$ and the membrane potential at the spike time. We can get:*

$$V\left(t_{emit}^{(l)}\right) + \sum_{v=0}^{t_i^{(l)}-t_{emit}^{(l)}-1} \psi_i^{(l)}(v) - \Delta_i^{(l)} = \theta_i^{(l)}. \tag{19}$$

*We denote $S(t) = \sum_{v=0}^{t-t_{emit}^{(l)}-1} \psi_i^{(l)}(v)$, then:*

$$S(t_i^{(l)}) = \theta_i^{(l)} + \Delta_i^{(l)} - V\left(t_{emit}^{(l)}\right). \tag{20}$$

*Then:*

$$t_i^{(l)} = S^{-1}\left( \theta_i^{(l)} + \Delta_i^{(l)} - V\left(t_{emit}^{(l)}\right) \right). \tag{21}$$

*According to the relationship between $x_i^{(l)}$ and $t_i^{(l)}$, we can get:*

$$x_i^{(l)} = \frac{1}{\tau_i^{(l)}} \left( t_{ref}^{(l)} - S^{-1}\left( \theta_i^{(l)} + \Delta_i^{(l)} - V\left(t_{emit}^{(l)}\right) \right) \right). \tag{22}$$

*let $W^{(l)} = (w_{ij}^{(l)})_{I \times I}$ is the weight matrix:*

$$x_i^{(l)} = f^{(l)}(W^{(l)}; x_1^{(l-1)}, ..., x_I^{(l-1)}) \tag{23}$$

$$= \frac{1}{\tau_i^{(l)}} \left( t_{ref}^{(l)} - S^{-1}\left( \theta_i^{(l)} + \Delta_i^{(l)} - \frac{1}{\tau_i^{(l)}} \sum_{j=1}^{I} \sum_{t=t_j^{(l-1)}}^{T_{emit}^{(l)}-1} w_{ij}^{(l)} \eta_{ij}^{(l)} \left( x_j^{(l-1)} \tau_j^{(l-1)} + t - t_{ref}^{(l-1)} \right) - T^{(l)} C_i^{(l)} \right) \right),$$

**proof 2** *Proof of Theorem 2:*
*Consider the potential change in the receiving stage.*

$$V_i\left(t_{emit}^{(l)}\right)$$

$$= \frac{1}{\tau_i^{(l-1)}} \sum_{t=t_{emit}^{(l-1)}}^{t_{end}^{(l-1)}-1} \sum_j w_{ij}^{(l)} \, \eta_{ij}^{(l)}\left(t - t_j^{(l-1)}\right) + C_i^{(l)}$$

$$= \frac{1}{\tau_i^{(l-1)}} \sum_j w_{ij}^{(l)} \sum_{u=0}^{t_{end}^{(l-1)}-t_j^{(l-1)}-1} \eta_{ij}^{(l)}(u) + d_i^{(l-1)} \cdot C_i^{(l)}$$

$$= \sum_j w_{ij}^{(l)} \frac{1}{\tau_i^{(l-1)}} \sum_{u=0}^{t_{end}^{(l-1)}-t_{ref}^{(l-1)}+\tau_i^{(l-1)} x_j^{(l-1)}-1} \eta_{ij}^{(l)}(u) + d_i^{(l-1)} \cdot C_i^{(l)}$$

$$= \sum_j w_{ij}^{(l)} \sum_{u=0}^{\tau_i^{(l-1)}\left(x_j^{(l-1)}-a_i^{(l-1)}\right)-1} \left(f_{ij}\left(\frac{u+1}{\tau_i^{(l-1)}} + a_i^{(l-1)}\right) - f_{ij}\left(\frac{u}{\tau_i^{(l-1)}} + a_i^{(l-1)}\right)\right) + d_i^{(l-1)} \cdot C_i^{(l)}$$

$$= \sum_j w_{ij}^{(l)} \left(f_{ij}(x_j^{(l-1)}) - f_{ij}(a_i^{(l-1)})\right) + d_i^{(l-1)} \cdot C_i^{(l)}$$

$$= \sum_j w_{ij}^{(l)} f_{ij}(x_j^{(l-1)}). \tag{24}$$

*where the second equation uses $u = t - t_j^{(l-1)}$; third equation uses $x_j^{(l-1)} \tau_i^{(l-1)} = t_{ref}^{(l-1)} - t_j^{(l-1)}$; fourth equation uses $a_i^{(l-1)} = \frac{t_{ref}^{(l-1)}-t_{end}^{(l-1)}}{\tau_i^{(l-1)}}$.*

*If the spike is emitted at time $t_i^{(l)} \in \{t_{emit}^{(l)}, t_{emit}^{(l)} + 1, \ldots, t_{end}^{(l)}\}$, i.e. the corresponding value $x_i^{(l)} \in [a_i^{(l)}, b_i^{(l)}]$. Then:*

$$\theta^{(l)} = V(t_{emit}^{(l)}) + \sum_{v=0}^{t_i^{(l)}-t_{emit}^{(l)}-1} \psi_i^{(l)}(v) - \Delta_i^{(l)}$$

$$= V(t_{emit}^{(l)}) + \sum_{v=0}^{t_{ref}^{(l)}-t_{emit}^{(l)}-\tau_i^{(l)} x_i^{(l)}-1} \left(h^{-1}(b_i^{(l)} - \frac{v}{\tau_i^{(l)}}) - h^{-1}(b_i^{(l)} - \frac{v+1}{\tau_i^{(l)}})\right) - \Delta_i^{(l)}$$

$$= V(t_{emit}^{(l)}) + \sum_{v=0}^{\tau_i^{(l)}(b_i^{(l)}-x_i^{(l)})-1} \left(h^{-1}(b_i^{(l)} - \frac{v}{\tau_i^{(l)}}) - h^{-1}(b_i^{(l)} - \frac{v+1}{\tau_i^{(l)}})\right) - \Delta_i^{(l)}$$

$$= V(t_{emit}^{(l)}) + h^{-1}(b_i^{(l)}) - h^{-1}(x_i^{(l)}) - \Delta_i^{(l)}. \tag{25}$$

*where the first equation uses $v = t_i^{(l)} - t_{emit}^{(l)}$, the second equation uses $x_i^{(l)} \tau_i^{(l)} = t_{ref}^{(l)} - t_i^{(l)}$, the third equation uses $b_i^{(l)} = \frac{t_{ref}^{(l)}-t_{emit}^{(l)}}{\tau_i^{(l)}}$.*

*Thus*

$$h^{-1}(x_i^{(l)}) = V(t_{emit}^{(l)}).$$

*which indicates that*

$$x_i^{(l)} = h(V(t_{emit}^{(l)})).$$

*If $h(V(t_{emit}^{(l)})) > b_i^{(l)}$, then $V(t_{emit}^{(l)}) > h^{-1}(b_i^{(l)}) = \theta_i^{(l)}$, which means that a spike is emitted once at $t_{emit}^{(l)}$, representing the value $\frac{t_{ref}^{(l)}-t_{emit}^{(l)}}{\tau_i^{(l)}} = b_i^{(l)}$.*

If $h(V(t_{emit}^{(l)})) < a_i^{(l)}$, then the potential at time $t_{end}^{(l)}$ is:

$$V(t_{emit}^{(l)}) + \sum_{v=0}^{T^{(l)}} \psi_i^{(l)}(v)$$

$$= V(t_{emit}^{(l)}) + \sum_{v=0}^{T^{(l)}} \left( h^{-1}(b_i^{(l)} - \frac{v}{\tau_i^{(l)}}) - h^{-1}(b_i^{(l)} - \frac{v+1}{\tau_i^{(l)}}) \right)$$

$$= V(t_{emit}^{(l)}) + h^{-1}(b_i^{(l)}) - h^{-1}(a_i^{(l)})$$

$$< h^{-1}(b_i^{(l)}) = \theta_i^{(l)}. \tag{26}$$

which means that there will be no spike, representing the value $a_i^{(l)}$.

**proof 3** *Proof of Lemma 2:*
*According to Theorem 4.3 in Zhao et al. (2025): in the continuous setting, if the spike is emitted at time $t_i^{(l)} \in [t_{emit}^{(l)}, t_{end}^{(l)}]$, i.e. the corresponding value $y_i^{(l)} \in [a_i^{(l)}, b_i^{(l)}]$. Then*

$$\theta_i^{(l)} = V(t_{emit}^{(l)}) + \int_0^{t_i^{(l)} - t_{emit}^{(l)}} \psi_i^{(l)}(v) \, dv \tag{27}$$

$$= V(t_{emit}^{(l)}) + \int_0^{t_{ref}^{(l)} - t_{emit}^{(l)} - \tau_i^{(l)} y_i^{(l)}} \frac{1}{\tau_i^{(l)}} (h^{-1})' \left( b_i^{(l)} - \frac{v}{\tau_i^{(l)}} \right) dv \tag{28}$$

$$= V(t_{emit}^{(l)}) - h^{-1} \left( b_i^{(l)} - \frac{v}{\tau_i^{(l)}} \right) \Big|_0^{\tau_i^{(l)}(b_i^{(l)} - y_i^{(l)})} \tag{29}$$

$$= V(t_{emit}^{(l)}) - h^{-1}(y_i^{(l)}) + h^{-1}(b_i^{(l)}). \tag{30}$$

*Because $\theta_i^{(l)} = h^{-1}(b_i^{(l)})$:*

$$h^{-1}(y_i^{(l)}) = V(t_{emit}^{(l)}), \tag{31}$$

*which indicates that:*

$$y_i^{(l)} = h\left( V(t_{emit}^{(l)}) \right). \tag{32}$$

*In the discrete setting, the spike is emitted at time $t_i^{(l)} \in \{t_{emit}^{(l)}, t_{emit}^{(l)} + 1, \ldots, t_{end}^{(l)}\}$, the corresponding value $x_i^{(l)} \in [a_i^{(l)}, b_i^{(l)}]$. Let $\Delta_i^{(l)} \geq 0$ is a compensation constant, which is actually the difference between the $\theta_i^{(l)}$ and the membrane potential at the spike time. The following equation satisfies:*

$$\theta_i^{(l)} = V(t_{emit}^{(l)}) + \int_0^{t_i^{(l)} - t_{emit}^{(l)}} \psi_i^{(l)}(v) \, dv - \Delta_i^{(l)} \tag{33}$$

$$= V(t_{emit}^{(l)}) + \int_0^{t_{ref}^{(l)} - t_{emit}^{(l)} - \tau^{(l)} x_i^{(l)}} \frac{1}{\tau_i^{(l)}} (h^{-1})' \left( b_i^{(l)} - \frac{v}{\tau_i^{(l)}} \right) dv - \Delta_i^{(l)} \tag{34}$$

$$= V(t_{emit}^{(l)}) - h^{-1} \left( b_i^{(l)} - \frac{v}{\tau_i^{(l)}} \right) \Big|_0^{\tau_i^{(l)}(b_i^{(l)} - x_i^{(l)})} - \Delta_i^{(l)} \tag{35}$$

$$= V(t_{emit}^{(l)}) - h^{-1}(x_i^{(l)}) + h^{-1}(b_i^{(l)}) - \Delta_i^{(l)}. \tag{36}$$

*Because $\theta_i^{(l)} = h^{-1}(b_i^{(l)})$:*

$$h^{-1}(x_i^{(l)}) = V(t_{emit}^{(l)}) - \Delta_i^{(l)}, \tag{37}$$

which indicates that:

$$x_i^{(l)} = h\left(V(t_{emit}^{(l)}) - \Delta_i^{(l)}\right). \tag{38}$$

The error of discrete coding in the continuous setting can be expressed as:

$$\left\|y_i^{(l)} - x_i^{(l)}\right\| = \left\|h\left(V(t_{emit}^{(l)})\right) - h\left(V(t_{emit}^{(l)}) - \Delta_i^{(l)}\right)\right\| \tag{39}$$

By the mean value theorem, we obtain:

$$\left\|y_i^{(l)} - x_i^{(l)}\right\| = \left\||h'(\xi)| \cdot \Delta_i^{(l)}\right\|, \tag{40}$$

where $\xi \in \left[V(t_{emit}^{(l)}) - \Delta_i^{(l)}, V(t_{emit}^{(l)})\right]$.

Furthermore, we examine $\Delta_i^{(l)}$ to provide a more in-depth analysis of the error. We assume that the spike firing time corresponding precisely to the ANN output is denoted as $[t]_i^{(l)}$. Based on the characteristics of TTFS encoding, it follows that:

$$t_i^{(l)} - 1 \leq [t]_i^{(l)} \leq t_i^{(l)}. \tag{41}$$

Then $\Delta_i^{(l)}$ can be represented as:

$$\Delta_i^{(l)} = \int_{[t]_i^{(l)} - t_{emit}^{(l)}}^{t_i^{(l)} - t_{emit}^{(l)}} \psi_i(s)\, ds = h^{-1}(t_i^{(l)} - t_{emit}^{(l)}) - h^{-1}([t]_i^{(l)} - t_{emit}^{(l)}). \tag{42}$$

By the mean value theorem, we obtain:

$$\left\|\Delta_i^{(l)}\right\| = \left\||(h^{-1})'(\hat{t}_i)| \cdot (t_i^{(l)} - [t]_i^{(l)})\right\|, \tag{43}$$

where $\hat{t}_i \in \left[[t]_i^{(l)} - t_{emit}^{(l)}, t_i^{(l)} - t_{emit}^{(l)}\right]$.

Then the error $\epsilon_i^{(l)}$ can be bounded by the following inequality:

$$\left\|y_i^{(l)} - x_i^{(l)}\right\| \leq |h'(\xi)| \cdot \left|(h^{-1})'(\hat{t}_i)\right| \cdot |[t]_i^{(l)} - t_i^{(l)}| \tag{44}$$

By the definition of clock precision: $\Delta t_{real} = t_{real}(t + 1) - t_{real}(t)$, where $t_{real}(t) = t \cdot \Delta t_{real}$, we obtain:

$$\left\|y_i^{(l)} - x_i^{(l)}\right\| \leq |h'(\xi)| \cdot \left|(h^{-1})'(\hat{t}_i)\right| \cdot \Delta t_{real}$$

$$= |h'(\xi)| \cdot \left|(h^{-1})'(\hat{t}_i)\right| \cdot \frac{\Omega}{t_{end}^{(l)} - t_{emit}^{(l)}}$$

$$= |h'(\xi)| \cdot \left|(h^{-1})'(\hat{t}_i)\right| \cdot \frac{\Omega}{T^{(l)}}$$

$$\leq \frac{G_1 G_2 \Omega}{T^{(l)}} \tag{45}$$

**proof 4** *Proof of Theorem 3:*
*For clipping error, according to Lemma 1, we can get:*

$$\mathcal{E}_c^{(l)} = \begin{cases} \left\|\alpha_i^{(l)} - b_i^{(l)}\right\| & if\, \alpha_i^{(l)} > b_i^{(l)} \\ 0 & if\, \alpha_i^{(l)} \in [a_i^{(l)}, b_i^{(l)}] \\ \left\|a_i^{(l)} - \alpha_i^{(l)}\right\| & if\, \alpha_i^{(l)} < a_i^{(l)}, \end{cases} \tag{46}$$

*We define the center of the output interval of SNN as:*

$$c_i^{(l)} = \frac{a_i^{(l)} + b_i^{(l)}}{2} \tag{47}$$

*The clipping error can then be restated as follows:*

$$\mathcal{E}_c^{(l)}(T^{(l)}) = \max\left(\left|\alpha_i^{(l)} - c_i^{(l)}\right| - \frac{T^{(l)}}{2\tau_i^{(l)}}, 0\right) \tag{48}$$

*We take the derivative of $T^{(l)}$ to get the sensitivity of the error $\mathcal{E}_c^{(l)}(T^{(l)})$ to $T^{(l)}$:*

$$\frac{d}{dt}\mathcal{E}_c^{(l)}(T^{(l)}) = \begin{cases} -\dfrac{1}{2\tau_i^{(l)}}, & \left|\alpha_i^{(l)} - c_i^{(l)}\right| > \dfrac{T^{(l)}}{2\tau_i^{(l)}}, \\ 0, & \left|\alpha_i^{(l)} - c_i^{(l)}\right| < \dfrac{T^{(l)}}{2\tau_i^{(l)}}. \end{cases} \tag{49}$$

*Once clipping occurs, increasing $T^{(l)}$ will reduce the error linearly with a constant slope of $-\frac{1}{2\tau_i^{(l)}}$; within the valid interval, the error is unaffected by $T^{(l)}$.*

*For quantization error, according to Lemma 2, we can get:*

$$\mathcal{E}_q^{(l)} \leq \frac{G_1 G_2 \Omega}{T^{(l)}}. \tag{50}$$

*For an L-layer network with I neurons in each layer, we can get:*

$$\mathcal{E} = \sum_{i=1}^{I}\sum_{l=1}^{L}\left(\mathcal{E}_c^{(l)} + \mathcal{E}_q^{(l)}\right)$$

$$\leq \sum_{i=1}^{I}\sum_{l=1}^{L}\left(\max\left(\left|\alpha_i^{(l)} - c_i^{(l)}\right| - \frac{T^{(l)}}{2\tau_i^{(l)}}, 0\right) + \frac{G_1 G_2 \Omega}{T^{(l)}}\right) \tag{51}$$

*Let $T = \min\left\{T^{(l)}\right\}_{l=1}^{L}$ and $\tau = \max\left\{\left\{\tau_i^{(l)}\right\}_{i=1}^{I}\right\}_{l=1}^{L}$:*

$$\mathcal{E} \leq LI \cdot \max\left(\left|\alpha_i^{(l)} - \frac{a_i^{(l)} + b_i^{(l)}}{2}\right| - \frac{T^{(l)}}{2\tau_i^{(l)}}, 0\right) + \frac{LI G_1 G_2 \Omega}{T} \tag{52}$$

**proof 5** *Proof of Corollary 1:*

**Input transform:** *The input of QANN at $i$-th neuron of $l$-th layer is $\mathbf{X}_i^{(l)} = \sum_j w_{ij}^{(l)} x_j^{(l-1)} \in [a_i^{(l-1)}, b_i^{(l-1)}]$. In order to approximate the input of QANN, based on Theorem 2, we set the kernel function $\eta_{ij}^{(l)}$ and $C_i^{(l)}$ as follows:*

$$\eta_{ij}^{(l)}(u) = \mathcal{H}\left(\frac{u}{\tau_i^{(l-1)}} + a_i^{(l-1)}\right),$$

$$C_i^{(l)} = \sum_j \frac{a_i^{(l-1)}}{d_i^{(l-1)}} w_{ij},$$

*Then, the membrane potential after reception is completed can be expressed as:*

$$V(T_{emit}) = \sum_j w_{ij}^{(l)} x_j^{(l-1)} = \mathbf{X}_i^{(l)}. \tag{53}$$

**Output transform:** *In order to approximate the output of QANN at $l$-th layer of $i$-th neuron: $\hat{\mathbf{X}}^{(l)} \in [a_i^{(l)}, b_i^{(l)}]$, based on Theorem 2, we set the kernel function $\psi_i^{(l)}$ and threshold as follows:*

$$\psi_i^{(l)}(v) = \frac{1}{\tau_i^{(l)}}, \quad \theta_i^{(l)} = b_i^{(l)} \tag{54}$$

*If the spike is emitted at time $t \in \{t_{emit}^{(l)}, t_{emit}^{(l)} + 1, \ldots, t_{end}^{(l)}\}$:*

$$X_i^{(l-1)} + \frac{1}{\tau_i^{(l)}} \cdot t \geq \theta_i^{(l)}. \tag{55}$$

*According to the definition of t, we can get:*

$$t = \left\lceil (\theta_i^{(l)} - \mathbf{X}_i^{(l)}) \tau_i^{(l)} \right\rceil \tag{56}$$

*According to the rounding range of t, we add the clip function to get:*

$$t = \text{clip}\left(\left\lceil (\theta_i^{(l)} - \mathbf{X}_i^{(l)}) \tau_i^{(l)} \right\rceil, t_{emit}^{(l)}, t_{end}^{(l)}\right) \tag{57}$$

*According to the relation between spike time and corresponding activation value and $\theta_i^{(l)} = b_i^{(l)}$:*

$$x_i^{(l)} = \frac{1}{\tau_i^{(l)}} \left(t_{ref}^{(l)} - \text{clip}(\left\lceil (b_i^{(l)} - \mathbf{X}_i^{(l)}) \tau_i^{(l)} \right\rceil, t_{emit}^{(l)}, t_{end}^{(l)})\right) \tag{58}$$

*According to $b_i^{(l)} = \frac{t_{ref}^{(l)} - t_{emit}^{(l)}}{\tau_i^{(l)}}$, we can get:*

$$x_i^{(l)} = \frac{1}{\tau_i^{(l)}} \left(t_{ref}^{(l)} - \text{clip}(\left\lceil t_{ref}^{(l)} - t_{emit}^{(l)} - \mathbf{X}_i^{(l)} \tau_i^{(l)} \right\rceil, t_{emit}^{(l)}, t_{end}^{(l)})\right) \tag{59}$$

*Based on the relationship between the ceiling function and the floor function, we can derive the following:*

$$x_i^{(l)} = \frac{1}{\tau_i^{(l)}} \text{clip}(\left\lfloor \mathbf{X}_i^{(l)} \tau_i^{(l)} \right\rfloor - t_{ref}^{(l)} + t_{emit}^{(l)}, -t_{end}^{(l)}, -t_{emit}^{(l)}) + \frac{1}{\tau_i^{(l)}} t_{ref}^{(l)} \tag{60}$$

*Based on the properties of the floor function, we can conclude that:*

$$x_i^{(l)} = \frac{1}{\tau_i^{(l)}} \text{clip}(\left\lfloor \mathbf{X}_i^{(l)} \tau_i^{(l)} \right\rfloor + t_{end}^{(l)} - t_{ref}^{(l)} + t_{emit}^{(l)}, 0, t_{end}^{(l)} - t_{emit}^{(l)}) + \frac{1}{\tau_i^{(l)}} \left(t_{ref}^{(l)} - t_{end}^{(l)}\right) \tag{61}$$

*Let $t_{emit}^{(l)} = 0$, $t_{end}^{(l)} = N$, $\tau_i^{(l)} = \frac{1}{\lambda_i^{(l-1)}}$, $t_{end}^{(l)} - t_{ref}^{(l)} = z^{(l)}$, we can get:*

$$x_i^{(l)} = \lambda_i^{(l)} \cdot \text{clip}(\left\lfloor \frac{\mathbf{X}_i^{(l)}}{\lambda_i^{(l)}} \right\rfloor + z^{(l)}, 0, N) - \lambda_i^{(l)} z^{(l)} = \hat{\mathbf{X}}^{(l)} \tag{62}$$

## E NONLINEAR OPERATIONS IN QC-A2S

**Corollary 2 (Construction of SiLU)** *A TTFS-based neuron can be made equivalent to a discrete SiLU function with through the following configuration:*

$$\eta_{ij}^{(l)}(u) = \mathbb{I}[u \geq 0] \cdot \tau_i^{(l-1)} \cdot \left(\left(\frac{u+1}{\tau_i^{(l-1)}} + a_i^{(l-1)}\right) \cdot \sigma\left(\frac{u+1}{\tau_i^{(l-1)}} + +a_i^{(l-1)}\right) - \left(\frac{u}{\tau_i^{(l-1)}} + +a_i^{(l-1)}\right) \cdot \sigma\left(\frac{u}{\tau_i^{(l-1)}} + a_i^{(l-1)}\right)\right)$$

$$C_i^{(l)} = \sum_j w_{ij}^{(l)} \frac{a_i^{(l-1)} \cdot \sigma(a_i^{(l-1)})}{d_i^{(l-1)}}, \; \psi_i^{(l)}(v) = \frac{1}{\tau_i^{(l)}}, \; \sigma(x) = \frac{1}{1 + e^{-x}}. \tag{63}$$

**Corollary 3 (Construction of GELU)** *A TTFS-based neuron can be made equivalent to a discrete GELU function with*

$$\eta_{ij}^{(l)}(u) = \mathbb{I}[u \geq 0] \cdot \tau_i^{(l-1)} \cdot \left(\left(\frac{u+1}{\tau_i^{(l-1)}} + a_i^{(l-1)}\right) \cdot \Phi\left(\frac{u+1}{\tau_i^{(l-1)}} + +a_i^{(l-1)}\right) - \left(\frac{u}{\tau_i^{(l-1)}} + +a_i^{(l-1)}\right) \cdot \Phi\left(\frac{u}{\tau_i^{(l-1)}} + a_i^{(l-1)}\right)\right)$$

$$\psi_i^{(l)}(v) = \frac{1}{\tau_i^{(l)}}, \; \theta_i^{(l)} = b_i^{(l)}, \; C_i^{(l)} = \sum_j w_{ij}^{(l)} \frac{a_i^{(l)} \Phi(a_i^{(l)})}{d_i^{(l-1)}}, \; \Phi(x) = \frac{1}{2}\left[1 + \text{erf}\left(\frac{x}{\sqrt{2}}\right)\right] \quad \text{erf}(x) = \frac{2}{\sqrt{\pi}} \int_0^x e^{-t^2}$$

**Corollary 4 (Construction of Softmax)** *The log-sum-exp of $I$ inputs $x_1, x_2, \cdots, x_I$, i.e.,*

$$\log \sum_{j=1}^{I} e^{x_j}, \tag{64}$$

*can be calculated in a single neuron with*

$$\eta_{ij}^{(l)}(u) = \tau_i^{(l-1)} \cdot \left( \exp\left( \frac{u+1}{\tau_i^{(l-1)}} + a_i^{(l-1)} \right) - \exp\left( \frac{u}{\tau_i^{(l-1)}} + a_i^{(l-1)} \right) \right). \tag{65}$$

$$C_i^{(l)} = \frac{I}{d_i^{(l-1)}} e^{a_i^{(l-1)}}, \; \psi_i^{(l)}(v) = \frac{1}{\tau_i^{(l)}} \exp\left( b_i^{(l)} - \frac{v}{\tau_i^{(l)}} \right). \tag{66}$$

*With the log-sum-exp neuron, we can obtain the softmax operator. We can calculate the logarithm of softmax, i.e.*

$$\log\left( \frac{e^{x_i}}{\sum_{j=1}^{I} e^{x_j}} \right) = x_i - \log \sum_{j=1}^{I} e^{x_j}, \tag{67}$$

*by subtracting the log-sum-exp from $x_i$. Finally, we can obtain the output after an exponential layer.*

**Corollary 5 (Construction of RMSNorm)** *RMSNorm is a normalization method widely used in LLaMA architecture, which is a linear operation. RMSNorm is defined as:*

$$\mathrm{RMSNorm}(x_i) = \frac{x_i}{\sqrt{\frac{1}{I} \sum_{i=1}^{I} x_i^2}} \cdot \gamma + \beta. \tag{68}$$

*We first can obtain the $\frac{1}{I} \sum_{i=1}^{I} x_i^2$ by a single neuron with*

$$\eta_{ij}^{(1)}(u) = \tau_{ij}^{(0)} \left[ \left( \frac{u+1}{\tau_i^{(0)}} + a_i^{(0)} \right)^2 - \left( \frac{u}{\tau_i^{(0)}} + a_i^{(0)} \right)^2 \right], \; C_i^{(1)} = \frac{(a_i^{(0)})^2}{I d_i^{(0)}}, \; w^{(1)} = \frac{1}{I} \tag{69}$$

$$\psi_i^{(1)}(v) = \frac{1}{\tau_i^{(1)}}, \; \theta_i^{(1)} = b_i^{(1)}. \tag{70}$$

*Then, we can get $\frac{1}{\sqrt{\frac{1}{I} \sum_{i=1}^{I} x_i^2}}$ with:*

$$\eta_{ij}^{(2)}(u) = \tau_{ij}^{(1)} \left[ \left( \frac{u+1}{\tau_i^{(1)}} + a_i^{(1)} \right)^{-\frac{1}{2}} - \left( \frac{u}{\tau_i^{(1)}} + a_i^{(1)} \right)^{-\frac{1}{2}} \right], \; C_i^{(2)} = \frac{1}{I d_i^{(1)} (a_i^{(1)})^{\frac{1}{2}}}, \tag{71}$$

$$w^{(2)} = 1 \; \psi_i^{(l)}(v) = \frac{1}{\tau_i^{(l)}}, \; \theta_i^{(l)} = b_i^{(l)}. \tag{72}$$

*Finally, multiply $x_i$ with $\frac{1}{\sqrt{\frac{1}{I} \sum_{i=1}^{I} x_i^2}}$.*

## F    USE OF LLMS

In this work, LLMs are employed solely for polishing or grammar checking text that is originally written by us.

# G  SUPPLEMENTARY MATERIALS FOR THE REBUTTAL

## G.1  SUPPLEMENTARY ALGORITHM FOR THE REBUTTAL

Algorithm 1 outlines the proposed QC-A2S pipeline for converting a vanilla LLM into a temporal-coding spiking LLM. The procedure begins by applying post-training quantization to obtain a well-quantized model. For each layer and each neuron, we retrieve its quantization parameters—including the quantization level, scaling factor, and zero-point—and use them to construct the corresponding TTFS-based spiking neuron with the emission time, ending time, and time constant defined accordingly. This mapping transforms all quantized linear computations into temporal spike representations. For layers containing nonlinear operators such as SwiGLU, Softmax, or LayerNorm, their original activations are replaced with the spiking counterparts derived in Appendix E, ensuring full compatibility with the temporal-coding SNN formulation. Through this layer-wise conversion, the entire LLM is transformed into a TTFS-based spiking model suitable for neuromorphic.

---

**Algorithm 1** Converting Vanilla LLM into Temporal-coding Spiking LLM with QC-A2S pipeline

---

1: **Input:** Vanilla LLM
2: **Output:** Temporal-coding Spiking LLM
3: **Obtaining well-quantized LLM by post-training quantization**
4: **Mapping from Quantized LLMs to Temporal-coding Spiking LLMs:**
5: **for** layer $l = 1, 2, ..., L$ **do**
6:     **for** neuron $i = 1, 2, ..., I$ **do**
7:         Get quantization parameters quantization level $N$, scale $\lambda_i^{(l)}$ and zero point $z^{(l)}$
8:         Construct TTFS-based spiking neural neurons as:
9:         $t_{\text{emit}}^{(l)} = 0,\ t_{\text{end}}^{(l)} = N,\ \tau_i^{(l)} = \frac{1}{\lambda_i^{(l)}},\ t_{\text{end}}^{(l)} - t_{\text{ref}}^{(l)} = z^{(l)}$
10:    **end for**
11:    **if** $l$ is SwiGLU, Softmax or RMSNorm **then**
12:       Replace the original nonlinear activation with spiking counterpart in the Appendix E.
13:    **end if**
14: **end for**

---

## G.2  SUPPLEMENTARY EXPERIMENTS FOR THE REBUTTAL

Figure 6 reports the layer-wise cumulative conversion error of the temporal-coding spiking LLM obtained with the QC-A2S framework on LLaMA-2-7B. The blue bars denote the conversion error at each layer, while the pink dashed curve shows the corresponding log-scaled error. The conversion error clearly accumulates as the layer index increases, which is consistent with Theorem 3, where the approximation error bound grows with the network depth $L$.

Tables 5 and 6 evaluate the generalization ability of our QC-A2S framework on non-LLaMA models. Table 5 reports accuracy for WinoGrande and ***acc_norm*** for HellaSwag, ArcC, ArcE, and PIQA on Qwen2-7B and Mistral-7B v0.3. Table 6 reports perplexity on C4, Pile, PTB, WikiText2, and Red-Pajama for the same models. The entries highlighted in gray correspond to the temporally encoded spiking LLM obtained with QC-A2S, while the non-shaded entries correspond to the equivalent quantized LLM. Across all benchmarks, the QC-A2S spiking models achieve accuracy and perplexity that are comparable to, and sometimes even better than, their quantized counterparts under the same precision configurations. These results demonstrate that our QC-A2S framework generalizes well across different LLM architectures and datasets.

Table 7 reports the performance of TTFS-Former on LLaMA models, where we measure accuracy on WinoGrande and ***acc_norm*** on HellaSwag, ArcC, ArcE, and PIQA. Across all model scales, TTFS-Former performs significantly worse than the FP16 baselines and our QC-A2S models, and the average accuracy further degrades as the time window T increases. This indicates that directly applying TTFS-Former to LLMs is not effective. We attribute this poor performance to two inherent limitations of TTFS-Former. First, TTFS-Former is built on a continuous-time TTFS assumption, where spike times are treated as real-valued variables. This formulation inherently requires a large

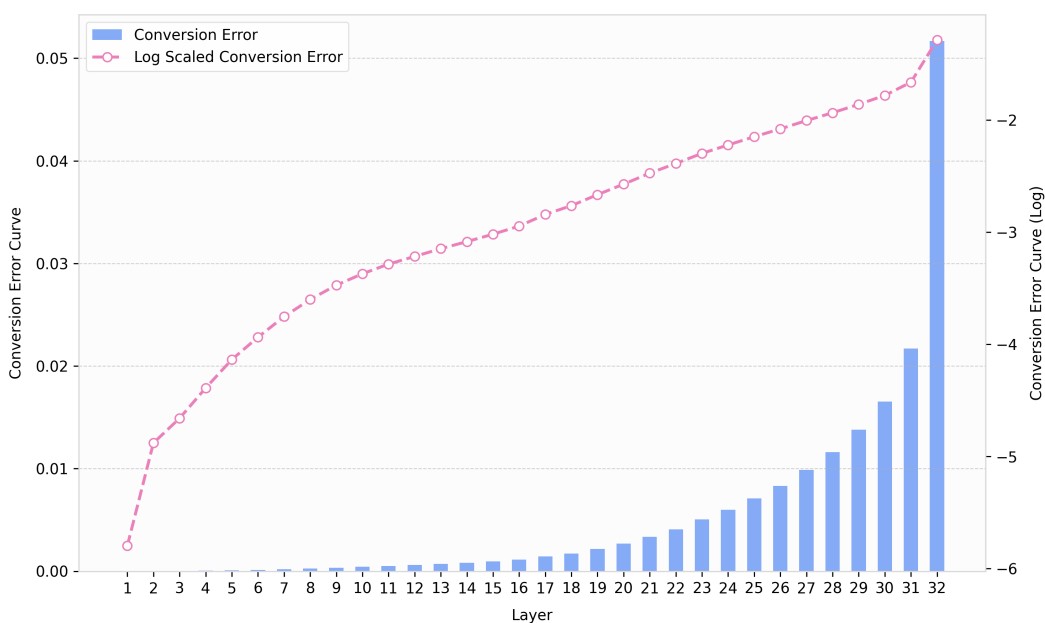

Figure 6: The layer-wise conversion error of temporal-coding spiking LLM (LLaMA-2-7B).

time window to approximate continuous dynamics, making medium-latency settings theoretically incompatible with its design. Second, TTFS-Former directly converts an ANN into an SNN without handling outliers, which are prevalent in LLM activations. These extreme values skew the spike-time distribution during conversion and lead to severe accuracy degradation—an intrinsic weakness of continuous-time ANN→SNN conversion pipelines. The results in Table 7 therefore highlight the necessity of our QC-A2S framework, which explicitly addresses these issues.

Table 5: We report *accuracy* for WinoGrande and *acc_norm* for HellaSwag, ArcC, ArcE, and PIQA on Qwen2-7B, and Mistral-7B v0.3 for demonstrating the generalization of our method. The entries highlighted in gray correspond to the temporally encoded spiking LLM, while the non-shaded entries correspond to the equivalent quantized LLM.

| Model | Precision | WinoGrande | HellaSwag | ArcC | ArcE | PIQA | Avg. |
|-------|-----------|------------|-----------|------|------|------|------|
| Qwen-2-7B | FP16 | 72.45 | 78.78 | 49.91 | 74.71 | 81.23 | 71.42 |
| | W4A4 | 68.19 | 74.67 | 49.66 | 76.05 | 79.00 | 69.51 |
| | T=16 | 68.59 | 74.56 | 48.12 | 75.13 | 78.13 | 68.91 |
| | W5A5 | 71.51 | 77.60 | 48.46 | 73.74 | 79.87 | 70.24 |
| | T=32 | 71.51 | 77.51 | 49.40 | 72.94 | 80.20 | 70.31 |
| | W6A6 | 71.51 | 78.68 | 51.19 | 75.51 | 80.36 | 71.45 |
| | T=64 | 71.82 | 78.41 | 51.45 | 76.09 | 81.18 | 71.79 |
| Mistral-7B v0.3 | T=16 | 73.88 | 80.44 | 52.30 | 78.20 | 82.26 | 73.42 |
| | W4A4 | 71.90 | 77.55 | 49.91 | 76.35 | 79.54 | 71.05 |
| | T=16 | 71.59 | 77.38 | 50.77 | 76.18 | 80.36 | 71.26 |
| | W5A5 | 72.53 | 79.74 | 52.90 | 79.25 | 80.85 | 73.05 |
| | T=32 | 73.09 | 79.63 | 52.82 | 78.79 | 81.23 | 73.11 |
| | W6A6 | 72.14 | 80.41 | 53.24 | 79.76 | 82.37 | 73.58 |
| | T=64 | 73.56 | 80.32 | 53.67 | 79.34 | 81.66 | 73.71 |

Table 6: We report **_perplexity_** for C4, Pile, PTB, WikiText2, and RedPajama on Qwen2-7B, and Mistral-7B v0.3 for demonstrating the generation quality of our method on other models. The entries highlighted in gray correspond to the temporally encoded spiking LLM, while the non-shaded entries correspond to the equivalent quantized LLM.

| Model | Precision | C4 | Pile | PTB | WikiText2 | RedPajama | Avg. |
|---|---|---|---|---|---|---|---|
| | FP16 | 9.90 | 5.53 | 12.22 | 7.14 | 8.51 | 8.66 |
| | W4A4 | 11.44 | 6.24 | 14.10 | 8.26 | 9.90 | 9.99 |
| | T=16 | 11.43 | 6.24 | 14.13 | 8.26 | 9.89 | 9.99 |
| Qwen-2-7B | W5A5 | 10.34 | 5.73 | 12.71 | 7.44 | 8.91 | 9.02 |
| | T=32 | 10.34 | 5.74 | 12.71 | 7.43 | 8.90 | 9.02 |
| | W6A6 | 10.05 | 5.60 | 12.35 | 7.23 | 8.64 | 8.78 |
| | T=64 | 10.05 | 5.60 | 12.36 | 7.23 | 8.64 | 8.78 |
| | T=16 | 7.84 | 4.46 | 37.28 | 5.32 | 6.00 | 12.18 |
| | W4A4 | 8.58 | 4.84 | 1353.43 | 5.93 | 6.65 | 275.89 |
| Mistral-7B v0.3 | T=16 | 8.58 | 4.84 | 1386.34 | 5.93 | 6.65 | 282.47 |
| | W5A5 | 8.12 | 4.61 | 1443.37 | 5.54 | 6.24 | 293.57 |
| | T=32 | 8.12 | 4.61 | 1416.23 | 5.54 | 6.24 | 288.15 |
| | W6A6 | 7.99 | 4.55 | 270.79 | 5.45 | 6.12 | 58.98 |
| | T=64 | 7.99 | 4.55 | 293.46 | 5.46 | 6.13 | 63.52 |

Table 7: We report extra ***accuracy*** for WinoGrande and ***acc_norm*** for HellaSwag, ArcC, ArcE, and PIQA on LLaMA models with TTFS-Former method.

| Model | Precision | WinoGrande | HellaSwag | ArcC | ArcE | PIQA | Avg. |
|-------|-----------|------------|-----------|------|------|------|------|
| 2-7B | FP16 | 69.22 | 76.00 | 46.33 | 74.62 | 79.11 | 69.06 |
|  | T=32 | 51.07 | 26.32 | 28.84 | 25.84 | 48.91 | 36.20 |
|  | T= 64 | 48.86 | 26.40 | 28.67 | 26.09 | 49.24 | 35.85 |
|  | T= 128 | 47.83 | 26.12 | 28.07 | 26.77 | 48.15 | 35.39 |
|  | T= 256 | 48.15 | 26.35 | 27.39 | 25.55 | 49.08 | 35.30 |
|  | T= 512 | 49.25 | 26.12 | 28.33 | 25.51 | 50.44 | 35.93 |
|  | T= 1024 | 49.88 | 26.07 | 29.52 | 25.84 | 49.46 | 36.15 |
|  | T= 2048 | 47.04 | 25.82 | 27.82 | 28.20 | 47.06 | 35.19 |
| 3-8B | FP16 | 72.69 | 79.19 | 53.41 | 77.69 | 80.79 | 72.75 |
|  | T= 32 | 49.88 | 26.40 | 26.54 | 24.71 | 51.58 | 35.82 |
|  | T= 64 | 50.59 | 26.19 | 26.11 | 24.49 | 50.16 | 35.51 |
|  | T= 128 | 51.78 | 25.88 | 26.37 | 25.08 | 50.65 | 35.95 |
|  | T= 256 | 50.20 | 26.48 | 26.79 | 25.46 | 51.80 | 36.15 |
|  | T= 512 | 51.38 | 26.35 | 25.17 | 24.41 | 51.36 | 35.73 |
|  | T= 1024 | 49.41 | 26.08 | 29.27 | 26.81 | 49.62 | 36.24 |
|  | T= 2048 | 48.78 | 26.73 | 26.96 | 25.88 | 51.85 | 36.04 |
| 2-13B | FP16 | 72.38 | 79.38 | 49.06 | 77.53 | 80.52 | 71.77 |
|  | T= 32 | 49.96 | 26.11 | 29.52 | 25.38 | 47.93 | 35.78 |
|  | T= 64 | 50.75 | 25.75 | 29.27 | 26.05 | 50.65 | 36.49 |
|  | T= 128 | 50.67 | 26.65 | 28.33 | 25.88 | 49.73 | 36.25 |
|  | T= 256 | 48.15 | 26.53 | 27.73 | 26.77 | 47.61 | 35.36 |
|  | T= 512 | 49.17 | 26.63 | 27.39 | 25.55 | 50.05 | 35.76 |
|  | T= 1024 | 51.78 | 26.16 | 27.82 | 26.98 | 49.35 | 36.42 |
|  | T= 2048 | 49.72 | 26.79 | 27.47 | 28.07 | 51.03 | 36.62 |
| 2-70B | FP16 | 77.98 | 83.82 | 57.42 | 80.98 | 82.70 | 76.58 |
|  | T= 32 | 48.15 | 26.40 | 28.24 | 25.00 | 49.02 | 35.36 |
|  | T= 64 | 48.54 | 26.15 | 30.03 | 26.30 | 51.03 | 36.41 |
|  | T= 128 | 50.20 | 25.85 | 28.67 | 26.81 | 50.44 | 36.39 |
|  | T= 256 | 50.99 | 25.76 | 28.84 | 25.59 | 50.11 | 36.26 |
|  | T= 512 | 49.64 | 26.17 | 28.58 | 26.39 | 50.00 | 36.16 |
|  | T= 1024 | 49.88 | 26.00 | 28.58 | 27.57 | 48.59 | 36.12 |
|  | T= 2048 | 50.12 | 26.36 | 28.24 | 26.30 | 50.22 | 36.25 |
| 3-70B | FP16 | 80.35 | 84.88 | 64.33 | 85.86 | 84.55 | 79.99 |
|  | T= 32 | 49.01 | 26.32 | 26.88 | 25.25 | 50.05 | 35.50 |
|  | T= 64 | 50.20 | 26.34 | 23.89 | 25.08 | 50.22 | 35.15 |

