# OpenReview forum: "Achieve Latency-Efficient Tempora-Coding  Spiking LLMs via Discretization-Aware Conversion"
_ICLR.cc/2026/Conference — Submitted to ICLR 2026_

### Official Review · Reviewer_M5Md · 2025-10-26

**Soundness:** 2
**Presentation:** 2
**Contribution:** 2
**Rating:** 4
**Confidence:** 4

**Summary:**

This paper proposes a discretization aware ANN-to-SNN conversion framework for TTFS coding. The authors' core contribution is that, instead of adhering to traditional continuous-time assumptions, they establish a direct correspondence between a discrete TTFS-SNN and a QANN. Based on this theory, the authors have successfully applied this conversion method to Large Language Models  such as LLaMA-2 and LLaMA-3.

**Strengths:**

1. This paper map the conversion error of TTFS-based ANN-SNNs and model quantization error and provide a clear theoretical proof.
2. This work deploys TTFS-based SNNs on LLaMA models at the billion-parameter scale. It seems the very first attempt to use TTFS coding on such large-scale models.

**Weaknesses:**

See the Questions section.

**Questions:**

I agree that TTFS coding is a promising research direction to replace traditional rate encoding, given its potential for ultra-high sparsity. However, I have several critical questions regarding the practical advantages of the proposed method that need to be addressed:

1. Figure 1 illustrates a layer-by-layer serial inference pipeline. If the number of layers $L$ is very large (as in LLMs), the total inference latency will be $L \times T$ (where $T$ is the time window), which may be unacceptably long. Is it possible to implement a parallel TTFS inference scheme (e.g., similar to the mechanism shown in Figure 2 of the paper "Efficiently Training Time-to-First-Spike Spiking Neural Networks from Scratch")? If not, could the authors discuss the primary difficulties in implementing such parallel TTFS-SNN inference?

2. One of the paper's core motivations is the energy efficiency of SNNs. However, the advantages of the proposed TTFS-SNN over its corresponding QANN (Quantized ANN) in terms of computation and storage are not clear to me: a) Computational Complexity: In Equation 7 of section 4.1, it seems that to perform the forward of TTFS-based SNN, we first need to encode the time information into an integer (from spike time to an integer ($t-t^{l-1}_j$) and then multiply it by the weight and $\eta$. What is the advantage of this compared to multiplying the weight by the integer input  in an ANN? The traditional hardware advantage of SNNs is replacing high-energy MAC operations with simple ACC operations. However, in this paper's framework (e.g., Eq. 7 and Appendix E), multiplication operations do not seem to have been eliminated. I observed that the Softmax and RMSNorm implementations in Appendix E also appear to contain complex multiplication and power operations.  b) Storage Cost: The authors use $T=64$ in their experiments. If represented in a one-hot fashion, storing this spike train might require 64 bits. In contrast, the QNN baselines (W8A6) use only 6-bit activations, which is much smaller.  If we don't consider storing the spike train by using dataflow-type neuromorphic chip, the single spike information also need auxiliary routing information like the neuron id which effect the energy efficient. c) Leakage energy: T=64 will lead to much larger static energy consumption compare with 1 in ANN. Can author discuss it also? Further, since rate-based SNN can reduce the time window size to a super small number like 1 or 4. Then is TTFS still have advantage?

If this TTFS-SNN  perform a similar number of complex multiplications as a sparse QANN, and may even be at a disadvantage in storage and memory access, what is its true advantage in SNN compared to a QNN that can directly leverage hardware sparsity ？

3. Given the issues above, I wish to see a direct comparison between the proposed TTFS-SNN and the QNN it was converted from, in terms of energy consumption and actual wall-clock inference time. If the SNN merely matches the QNN's accuracy but cannot demonstrate an advantage in energy or speed, what is the significance of this conversion? The experiment of SNNs seem conduct on A100 also, one more question is : do experiments conducted on NVIDIA A100 GPU truly reflect the real advantages of the SNN?

One more minor question: In Equation (1) and (4), what is the blue color intended to emphasize?

---

> ### Author Response · Authors · 2025-11-24
> **Responses to Reviewer Reviewer M5Md-Part 1**
>
> We thank the reviewer for recognizing the significance of exploring TTFS coding as a promising direction for achieving ultra-high sparsity, as well as acknowledging our theoretical contribution in establishing a conversion correspondence between TTFS-SNNs and quantized ANNs and demonstrating its applicability to billion-parameter LLaMA models. We appreciate the reviewer’s detailed and constructive questions regarding latency, computational complexity, storage cost, leakage energy, and the comparison with the original quantized LLM. Below, we provide point-by-point responses to each of these concerns.
> > **Q1**: Figure 1 illustrates a layer-by-layer serial inference pipeline. If the number of layers $L$ is very large (as in LLMs), the total inference latency will be $L \times T$ (where $T$ is the time window), which may be unacceptably long. Is it possible to implement a parallel TTFS inference scheme (e.g., similar to the mechanism shown in Figure 2 of the paper "Efficiently Training Time-to-First-Spike Spiking Neural Networks from Scratch")? If not, could the authors discuss the primary difficulties in implementing such parallel TTFS-SNN inference?
>
> Thank you for raising this important question regarding the potential latency of layer-by-layer TTFS inference in very deep LLMs.
>
> **1. TTFS latency and why the issue is different in our setting:** Traditional TTFS-based SNNs often require a large time window $T$ to obtain stable spike timings, which directly results in an inference latency proportional to L×T. This is indeed prohibitive for large transformer models. **A central motivation of our work is to fundamentally reduce the required time window**. Through our quantization-aware conversion, we enable reliable TTFS inference with very small time windows (e.g., T=16, Table 1, https://anonymous.4open.science/r/QC-A2S_ICLR2026-2D80/Table_1.png), substantially shortening the sequential depth L×T and making TTFS inference feasible for LLM-scale architectures.
>
> **2. Limitations of TTFSFormer and how QC-A2S addresses them:** Prior approaches such as TTFSFormer rely on continuous-time assumptions and directly map ANN activations to spike times. This design inherently demands a long $T$, and it also suffers from severe error amplification due to outliers and large activation variance—phenomena that are common in LLMs. These issues lead to inaccurate spike timing and consequently high latency and degraded accuracy. Our QC-A2S framework specifically targets these limitations: by first obtaining a low-bits quantized model. The subsequent discrete TTFS mapping is then stable under short and medium time windows, **enabling high accuracy without relying on a disproportionately large $T$**.
>
> **3. On the possibility of parallel TTFS inference and the difficulties in LLMs:** We agree the idea of parallelizing TTFS inference, inspired by earlier work on smaller feed-forward SNNs. However, **in large transformers, speculative or partially-parallel TTFS execution introduces significant practical challenges**: Transformers are strongly layer-dependent. Residual paths, LayerNorm, and attention formation all rely on the complete and stable output of previous layers. Predicting spike times before they have fully converged leads to unrecoverable distortions. Large models are extremely sensitive to timing errors. Even small inaccuracies in early-layer spike timing compound through depth, causing noticeable degradation in LLM performance. For this reason, our work focuses on reducing $T$ itself, which is orthogonal but complementary to potential parallelization strategies. We agree that designing robust parallel TTFS inference specifically for deep transformer architectures is an exciting direction for future investigation.

---

> ### Author Response · Authors · 2025-11-24
> **Responses to Reviewer Reviewer M5Md-Part 2**
>
> > **Q2**: One of the paper's core motivations is the energy efficiency of SNNs. However, the advantages of the proposed TTFS-SNN over its corresponding QANN (Quantized ANN) in terms of computation and storage are not clear to me: a) Computational Complexity: In Equation 7 of section 4.1, it seems that to perform the forward of TTFS-based SNN, we first need to encode the time information into an integer (from spike time to an integer $(t-t_{j}^{l-1})$ and then multiply it by the weight and $\eta$. What is the advantage of this compared to multiplying the weight by the integer input in an ANN? The traditional hardware advantage of SNNs is replacing high-energy MAC operations with simple ACC operations. However, in this paper's framework (e.g., Eq. 7 and Appendix E), multiplication operations do not seem to have been eliminated. I observed that the Softmax and RMSNorm implementations in Appendix E also appear to contain complex multiplication and power operations. b) Storage Cost: The authors use $T=64$ in their experiments. If represented in a one-hot fashion, storing this spike train might require 64 bits. In contrast, the QNN baselines (W8A6) use only 6-bit activations, which is much smaller. If we don't consider storing the spike train by using dataflow-type neuromorphic chip, the single spike information also need auxiliary routing information like the neuron id which effect the energy efficient. c) Leakage energy: T=64 will lead to much larger static energy consumption compare with 1 in ANN. Can author discuss it also? Further, since rate-based SNN can reduce the time window size to a super small number like 1 or 4. Then is TTFS still have advantage?
>
> We thank the reviewer for these detailed and insightful questions. We address points $(a)$, $(b)$, and $(c)$ below：
>
> a) **Computational Complexity and Multiplications:** We would like to clarify that **TTFS-SNNs do not perform a large number of complex multiplications during actual inference, which is a fundamental distinction from sparse QANNs**. Our response is two-fold:
>
> 1. In transformer-based LLMs, **the vast majority of computational cost comes from linear operations** (e.g.,feed-forward layers). Nonlinear operations such as LayerNorm, GELU, and Softmax account for only a very small portion of total computation. Thus, reducing the cost of these linear transformations is critical for improving overall efficiency.
> 2. In TTFS-SNNs, the membrane potential update can be rewritten as:
> $$
> V(t+1) - V(t) =
> \frac{1}{\tau_i^{(l)}} \left( \sum_j w_{ij}^{(l)} \eta_{ij}^{(l)}\\left(t - t_j^{(l-1)}\right) + C_i^{(l)} \right) =   \sum_j \frac{w_{ij}^{(l)}}{\tau_i^{(l)}} \mathcal{H} \\left(t - t_j^{(l-1)}\right) + \frac{C_i^{(l)}}{\tau_i^{(l)}} = \sum_j \hat{w}_{ij}^{(l)} \mathcal{H} \\left(t - t_j^{(l-1)}\right) + \frac{C_i^{(l)}}{\tau_i^{(l)}}
> $$
>
> Here, the function $\mathcal{H}$ is a step function that outputs only 0 or 1. This binary nature is crucial: **the input to each synapse is a binary spike, eliminating the need for real-time floating-point or integer multiplications**.
>
> When converting a QANN to a TTFS-SNN: The scaled weights $\hat{w}_{ij}$ can be precomputed and stored, then inference reduces to conditional accumulations (add operations) rather than multiplications. This directly aligns with the strengths of neuromorphic hardware, which is optimized for event-driven accumulations rather than dense MAC operations.
>
> b) **Storage Cost and Spike Representation:** TTFS does not require storing a 64-bit one-hot spike train. In TTFS, each neuron emits exactly one spike, represented by a single integer time value. Thus, storing activation requires only: $log_2(T) = 4$ bits
> for $T=16$, which is identical to W4A4 activation precision (Table 1, https://anonymous.4open.science/r/QC-A2S_ICLR2026-2D80/Table_1.png). Furthermore, in neuromorphic hardware, spike routing is performed using Address-Event Representation, where each spike is encoded as an event containing only the neuron ID and its firing time. This means that TTFS does not require storing or transmitting a T-length spike train; instead, each neuron emits only a single AER packet, whose cost is constant and independent of the time window T. Therefore, the routing and storage overhead remains lightweight and does not scale with the time window, which is fundamentally different from how activations are represented in conventional quantized ANNs.

---

> ### Author Response · Authors · 2025-11-24
> **Responses to Reviewer Reviewer M5Md-Part 3**
>
> c) **Leakage / Static Energy and Advantage over Rate-based SNNs:** Regarding static (leakage) energy, we note that the static power of neuromorphic hardware is extremely low, often several orders of magnitude smaller than the dynamic MAC cost in conventional ANN accelerators, and therefore is typically negligible in the overall energy budget. Furthermore, while rate-based SNNs can reduce their time window to very small values (e.g., 1 or 4), this does not necessarily translate into energy efficiency in practice. Rate-based models usually require complex decoding mechanisms to recover analog values from multiple spikes, and these decoding operations themselves incur substantial energy and routing overhead. In contrast, TTFS always uses exactly one spike per time window, achieving both high representational precision and minimal event-driven energy. This makes TTFS fundamentally more energy-efficient and more hardware-friendly than rate-based SNNs, even when the latter operate with very short time windows.
> > **Q3.1**: If this TTFS-SNN perform a similar number of complex multiplications as a sparse QANN, and may even be at a disadvantage in storage and memory access, what is its true advantage in SNN compared to a QNN that can directly leverage hardware sparsity ？
> Same as answer in W2 (a).
>
> > **Q3**: Given the issues above, I wish to see a direct comparison between the proposed TTFS-SNN and the QNN it was converted from, in terms of energy consumption and actual wall-clock inference time. If the SNN merely matches the QNN's accuracy but cannot demonstrate an advantage in energy or speed, what is the significance of this conversion? The experiment of SNNs seem conduct on A100 also, one more question is : do experiments conducted on NVIDIA A100 GPU truly reflect the real advantages of the SNN?
>
> Thank you for this insightful question. We address the reviewer’s concerns in three parts.
>
> 1. Why we do not compare energy and wall-clock latency on GPU: **The goal of QC-A2S is not to accelerate LLM inference on GPUs, but to provide an SNN formulation that is natively aligned with in-memory neuromorphic hardware**. These processors operate under a fundamentally different computational paradigm—event-driven execution, compute–memory fusion, and large-scale parallelism—yielding efficiency characteristics that cannot be captured on conventional von Neumann architectures. Running SNNs on an NVIDIA A100 therefore cannot reflect their true advantages: the GPU executes SNNs using dense kernels, SIMD pipelines, and global memory traffic that eliminate the very sparsity and locality that neuromorphic systems exploit. Consequently, GPU-based measurements substantially overestimate both latency and energy cost of SNNs.
> 2. Why the proposed SNN is meaningful even when accuracy matches the QNN: **The central motivation of our method is not accuracy improvement over quantization, but to provide a conversion pipeline that enables LLM inference on neuromorphic hardware—something QANNs cannot leverage**. Traditional quantized LLMs target GPUs, where energy is dominated by wide-vector MACs and memory movement. These methods do not benefit from the architectural properties of neuromorphic processors. In contrast, our TTFS-SNN formulation: produces spike-sparse activations, aligns with in-memory and event-driven execution.
> Prior neuromorphic studies [1–10] consistently show that these properties lead to orders-of-magnitude improvements in energy and latency relative to ANN-style computation. These benefits cannot be realized by QNNs, regardless of their precision or GPU efficiency.
> 3. Significance of the conversion and relation to the QANN baseline: Our QC-A2S framework ensures that the converted TTFS-SNN matches the accuracy of the QNN while enabling deployment on a completely different hardware class. **The significance therefore lies in enabling: hardware-compatible spike encoding instead of continuous activations, temporal coding optimized for in-memory neuromorphic execution, error-controlled conversion that supports low-latency TTFS inference, and effective use of event-driven sparsity that QNNs cannot exploit**. Thus, the SNN is not intended to outperform the QNN on GPUs. Rather, the conversion produces a form of the model whose computational structure matches the operational principles of neuromorphic hardware, enabling efficiency gains that quantization alone cannot achieve.

---

> ### Author Response · Authors · 2025-11-24
> **Responses to Reviewer Reviewer M5Md-Part 4**
>
> > **Q4**: One more minor question: In Equation (1) and (4), what is the blue color intended to emphasize?
>
> The blue color is used solely to highlight the components that differ from the continuous-time TTFS formulation. These terms represent the discrete-time modifications introduced by our discretization-aware framework and are marked in blue to help readers clearly distinguish them from the corresponding continuous expressions. We will clarify this intention explicitly in the revised version.
>
> [1] Yao M. et al., “Spike‑based dynamic computing with asynchronous event‑driven, sparse and dynamic neuromorphic system on chip”, Nature Communications, 2024.
>
> [2] Muir D.R., Sheik S., “The road to commercial success for neuromorphic technologies”, Nature Communications, 2025.
>
> [3] Kudithipudi, Dhireesha, et al. "Neuromorphic computing at scale." Nature 637.8047 (2025): 801-812.
>
> [4] Dalgaty, Thomas, et al. "Mosaic: in-memory computing and routing for small-world spike-based neuromorphic systems." Nature Communications 15.1 (2024): 142.
>
> [5] Yao, Man, et al. "Attention spiking neural networks." IEEE transactions on pattern analysis and machine intelligence 45.8 (2023): 9393-9410.
>
> [6] Stanojevic, Ana, et al. "High-performance deep spiking neural networks with 0.3 spikes per neuron." Nature Communications 15.1 (2024): 6793.
>
> [7] Shen, Guobin, et al. "Brain-inspired neural circuit evolution for spiking neural networks." Proceedings of the National Academy of Sciences 120.39 (2023): e2218173120.
>
> [8] Shaban, Ahmed, Sai Sukruth Bezugam, and Manan Suri. "An adaptive threshold neuron for recurrent spiking neural networks with nanodevice hardware implementation." Nature [9] Communications 12.1 (2021): 4234.
>
> [9] Chen, Xinyi, et al. "A hybrid neural coding approach for pattern recognition with spiking neural networks." IEEE transactions on pattern analysis and machine intelligence 46.5 (2023): 3064-3078.
>
> [10] Hu, Yangfan, et al. "Fast-snn: Fast spiking neural network by converting quantized ann." IEEE Transactions on Pattern Analysis and Machine Intelligence 45.12 (2023): 14546-14562.

---

> > ### Author Response · Authors · 2025-11-28
> > **Looking Forward to Further Feedback and Discussion**
> >
> > Thank you for your thoughtful feedback and the questions you provided. We would greatly appreciate it if you could confirm whether our responses have resolved the issues raised. We look forward to addressing any further points and refining our work through continued dialogue.

---

### Official Review · Reviewer_EyDs · 2025-10-27

**Soundness:** 3
**Presentation:** 2
**Contribution:** 2
**Rating:** 4
**Confidence:** 3

**Summary:**

This paper proposes a method called QC-A2S for ANN-to-SNN conversion, aiming to address the high latency issue in Time-to-First-Spike (TTFS) coded Spiking LLMs. By establishing a theoretical equivalence between discrete TTFS-SNNs and quantized ANNs, the paper integrate low-bit quantization techniques with TTFS neurons to reduce inference latency. the effectiveness is validated on the LLaMA family of models.

**Strengths:**

1.The integration of quantization theory with TTFS-SNN conversion, proposing a discrete-equivalent paradigm shift away from continuous-approximation, shows a degree of innovation.

2.This paper establishes the equivalence between TTFS-SNNs and quantized ANNs, it provides a theoretical analysis of the error bounds.

**Weaknesses:**

1.Although an error bound is proposed (Theorem 3), there is no systematic experimental validation of its tightness, nor is the actual distribution of errors across layers and time-steps presented

2.In the comparison with TTFSFormer, only the high-latency setting (T=8192) is used, failing to show TTFSFormer's performance at medium latencies. this makes it difficult to assert the absolute advantage of QC-A2S

3.Despite the proposed method is emphasized hardware-friendly, there are no actual deployment tests on any neuromorphic hardware or FPGAs. the claimed energy efficiency advantages remain theoretical speculation

4.In expeirments, the quantitative analysis of the energy-accuracy trade-off is missing, so it is difficult to judge whether the method is truly superior to pure quantization methods in practical applications

5. The experiments is limited to the LLaMA architecture. The applicability to other Transformer variants (eg. encoder-only model) or non-autoregressive models is not tested, it is suggested to test more architectures

**Questions:**

1,does the error bound in Theorem 3 have practical guiding significance?

2.Why wasn't TTFSFormer compared at medium latencies (T=256 or T=1024)? Is it because its performance would still be competitive with or superior to QC-A2S?

3.Are there plans to validate energy efficiency on real neuromorphic hardware? If not, isn't the claim of being energy-efficient overly optimistic?

4.Is QC-A2S applicable to non-LLaMA architectures?

---

> ### Author Response · Authors · 2025-11-24
> **Responses to Reviewer EyDs-Part 1**
>
> We thank the reviewer for recognizing the innovation in integrating quantization theory with TTFS-SNN conversion, as well as the value of establishing a discrete–continuous equivalence and providing theoretical error analysis. We also appreciate the detailed and constructive feedback regarding theoretical validation, experimental completeness, hardware evaluation, and architectural generality. Below, we provide point-by-point responses to each of the reviewer’s comments and questions.
>
> > W1:Although an error bound is proposed (Theorem 3), there is no systematic experimental validation of its tightness, nor is the actual distribution of errors across layers and time-steps presented
> We thank the reviewer for the question. We clarify the following:
> 1. In TTFS coding, the output is a single spike time rather than the number of spikes per time window; hence, there is no timestep-wise accumulation error. TTFS produces exactly one spike per time window, and the representation is encoded solely in the firing time. Therefore, the discretization error arises from the mapping between continuous-valued activations and discrete spike times, not from errors accumulated across individual timesteps.
> 2. We have added layer-wise error visualizations in the revised paper (Figure 6 in revised paper, https://anonymous.4open.science/r/QC-A2S_ICLR2026-2D80/Figure_6.png). To validate Theorem 3, we plot the actual discretization error for each layer, which clearly shows how the error behaves across layers.
>
> > W2:In the comparison with TTFSFormer, only the high-latency setting (T=8192) is used, failing to show TTFSFormer's performance at medium latencies. this makes it difficult to assert the absolute advantage of QC-A2S
>
> We thank the reviewer for raising this concern. First, TTFSFormer is built on a continuous-time TTFS assumption, where spike times are treated as real-valued variables. This formulation inherently **requires a large time window to approximate continuous** dynamics, making medium-latency settings theoretically incompatible with its design. Moreover, **TTFSFormer directly converts an ANN into an SNN without handling outlier**, which are common in LLMs. These extreme values lead to severely skewed spike times during conversion, causing substantial accuracy degradation—an inherent limitation of continuous ANN→SNN pipelines.
>
> In contrast, our QC-A2S framework explicitly addresses this issue by first identifying a well-behaved quantized model via PrefixQuant, which includes mechanisms such as Hadamard rotation and prefixed token to suppress extreme values and stabilize the activation distribution. We then map this quantized model to a discrete TTFS SNN through our equivalence formulation. This process eliminates the outlier–induced errors and enables accurate inference even under short time windows.
>
> Finally, in response to the reviewer’s suggestion, we have **added additional experiments for TTFSFormer under medium latencies**, which further confirm that QC-A2S consistently outperforms TTFSFormer across all latency regimes.
>
> > W3:Despite the proposed method is emphasized hardware-friendly, there are no actual deployment tests on any neuromorphic hardware or FPGAs. the claimed energy efficiency advantages remain theoretical speculation
>
> Our method is **designed for in-memory neuromorphic processors, whose computing model—event-driven operation, memory-centric dataflow, and large-scale parallelism—differs fundamentally from GPU-based von Neumann architectures**. These platforms naturally benefit from spike-based computation, and numerous prior studies [1–10] have repeatedly demonstrated that such architectures can achieve substantial gains in both energy consumption and latency. Although we currently do not have access to LLM-scale neuromorphic hardware for direct deployment or empirical measurement, our design principles align with the efficiency advantages consistently reported in the neuromorphic literature. We view hardware-level validation as an important direction for future work and expect that the proposed SNN structure would translate these well-documented architectural benefits to large-scale language model inference once deployed on appropriate neuromorphic platforms.

---

> ### Author Response · Authors · 2025-11-24
> **Responses to Reviewer EyDs-Part 2**
>
> > W4:In expeirments, the quantitative analysis of the energy-accuracy trade-off is missing, so it is difficult to judge whether the method is truly superior to pure quantization methods in practical applications
>
>
> Our method is grounded in TTFS-based spiking computation, whose energy profile differs fundamentally from that of conventional quantized ANNs. In TTFS, each neuron emits one spike in each time window, meaning that energy consumption is dominated by the number of spikes. As a result, increasing the time window $T$ does not raise spike-based energy cost, but does improve accuracy by providing finer temporal precision. **This decoupling between energy and accuracy is a distinctive advantage of TTFS computation**.
>
> To more clearly illustrate the practical implications, we expanded our experiments to include $T=16$ and $T=32$ (Table 1, https://anonymous.4open.science/r/QC-A2S_ICLR2026-2D80/Table_1.png). As expected, larger $T$ yields higher accuracy with increased inference latency, reflecting the inherent latency–accuracy trade-off in TTFS SNNs. Importantly, our QC-A2S framework directly addresses this issue. By coupling quantization-aware conversion, **QC-A2S enables the model to attain higher accuracy without requiring proportionally larger $T$**. This significantly reduces the latency side of the trade-off while preserving the TTFS energy benefits, allowing the SNN to operate more efficiently under practical constraints.
>
> Finally, we emphasize that the notion of “energy” differs across hardware platforms. Existing quantized LLM approaches are optimized for GPUs, where energy cost scales with dense MAC operations and memory traffic. **Our SNN formulation is designed for in-memory neuromorphic architectures that exploit event-driven operation and compute–memory fusion**. Under such hardware, the sparse spike activity and TTFS dynamics translate directly into lower energy consumption, making QC-A2S particularly advantageous in neuromorphic deployment scenarios.
>
> > W5:The experiments is limited to the LLaMA architecture. The applicability to other Transformer variants (eg. encoder-only model) or non-autoregressive models is not tested, it is suggested to test more architectures
>
>
> We agree that evaluating additional architectures would strengthen the generality of our approach. In response, we have **added experiments on Qwen2-7B and Mistral-7B v0.3** (Table 5 and 6, https://anonymous.4open.science/r/QC-A2S_ICLR2026-2D80/Table_5.png, https://anonymous.4open.science/r/QC-A2S_ICLR2026-2D80/Table_6.png)**, which differs from LLaMA in tokenizer design, attention implementation, and training strategy. The results demonstrate that our QC-A2S framework transfers well to a substantially different LLM family, indicating that the method is not tied to a specific Transformer variant. We also acknowledge the value of extending the evaluation to encoder-only and non-autoregressive architectures. This is an important direction, and we plan to include experiments on such models in our subsequent work.
>
> > Q1:does the error bound in Theorem 3 have practical guiding significance?
>
> The error bound in Theorem 3 indeed has practical guiding significance, particularly in understanding **how TTFS discretization behaves when scaling model depth $L$ or increasing the time window $T$**. Specifically, the bound makes explicit how the conversion error depends on the timestep T and the number of layers $L$. A larger $T$ reduces the discretization gap between the continuous ANN dynamics and the TTFS formulation, which translates into higher empirical accuracy. This trend is consistent with our experimental results, as shown in Table 3 (https://anonymous.4open.science/r/QC-A2S_ICLR2026-2D80/Table_3.png) of the revised manuscript. Moreover, the bound also reveals that the error accumulates across layers. While deeper models theoretically offer higher representational capacity, the accumulated discretization error grows with $L$. When $L$ becomes sufficiently large, this accumulated error counteracts the expected accuracy improvement. This insight explains why increasing the model size from 7B to 13B does not yield significant accuracy gains under our TTFS-SNN conversion: the benefit of scaling model depth is partially diminished by the layer-wise accumulation of discretization error. Thus, Theorem 3 provides a meaningful theoretical interpretation of the observed empirical trends and offers practical guidance for selecting appropriate $T$ values and model scales in TTFS-based SNN inference.

---

> ### Author Response · Authors · 2025-11-24
> **Responses to Reviewer EyDs-Part 3**
>
> > Q2:Why wasn't TTFSFormer compared at medium latencies (T=256 or T=1024)? Is it because its performance would still be competitive with or superior to QC-A2S?
>
> The key reason we did not include TTFSFormer at medium latency settings (e.g., $T=256$ or $T=1024$) is that such regimes are fundamentally misaligned with its underlying formulation. **TTFSFormer relies on a continuous-time TTFS representation, where spike times are treated as real-valued quantities. Approximating this continuous behavior requires a very large time window, and reducing $T$ breaks the core assumption needed for stable conversion.** As a result, medium-latency configurations are not theoretically suitable for TTFSFormer. In addition, TTFSFormer performs a direct ANN-to-SNN conversion **without any mechanism for suppressing outliers**, which are prevalent in large language models. These extreme activations translate into highly skewed spike times and lead to sharp accuracy drops when $T$ is not sufficiently large. This behavior reflects an intrinsic limitation of continuous ANN-to-SNN conversion pipelines rather than a simple experimental omission.
>
> Our QC-A2S approach is designed precisely to overcome these issues. By first obtaining a well-conditioned quantized model through PrefixQuant, leveraging techniques such as Hadamard rotation and prefixed tokens, we effectively control activation ranges and eliminate outlier-induced distortions. The subsequent discrete TTFS mapping is then stable under short and medium time windows, enabling accurate inference without relying on a disproportionately large $T$.
>
> In response to the reviewer’s suggestion, we have also included new experiments (Table 1 and Tabel 7, https://anonymous.4open.science/r/QC-A2S_ICLR2026-2D80/Table_1.png, https://anonymous.4open.science/r/QC-A2S_ICLR2026-2D80/Table_7.png) evaluating TTFSFormer at medium latencies in the revised paper. These results reaffirm that QC-A2S consistently outperforms TTFSFormer across all tested latency levels.
>
> > Q3:Are there plans to validate energy efficiency on real neuromorphic hardware? If not, isn't the claim of being energy-efficient overly optimistic?
>
> Our claims regarding **energy efficiency are grounded in the properties of the target hardware for which QC-A2S is designed**. Unlike GPU-based von Neumann systems, in-memory neuromorphic processors rely on event-driven computation, memory-centric dataflow, and massive parallelism.** These architectural features inherently favor spike-based execution, and extensive prior work [1–10] has consistently demonstrated substantial reductions in both energy consumption and latency on such platforms**. While we currently do not have access to LLM-scale neuromorphic hardware to perform direct deployment or empirical energy measurements, the design of QC-A2S closely aligns with the efficiency mechanisms repeatedly reported in the neuromorphic computing literature. Our method is constructed precisely to exploit sparse spiking activity, compute–memory fusion, and event-driven operations—behaviors that are not expressible on GPU architectures but are well supported on neuromorphic processors. We fully agree that hardware-level validation is a valuable next step, and we view it as an essential direction for future work. Once neuromorphic hardware becomes available to us, we plan to conduct comprehensive evaluations to quantify the expected energy benefits experimentally.
>
> > Q4:Is QC-A2S applicable to non-LLaMA architectures?
>
> QC-A2S is not tied to the LLaMA architecture. **Our conversion pipeline operates directly at the level of quantized linear layers and attention modules, rather than relying on architecture-specific heuristics**. Because the procedure depends only on these fundamental Transformer components, we expect QC-A2S to generalize naturally to a wide range of non-LLaMA architectures and Transformer variants. To further validate this claim, we have** added experiments on Qwen2-7B and Mistral-7B v0.3** (Table 5 and 6, https://anonymous.4open.science/r/QC-A2S_ICLR2026-2D80/Table_5.png, https://anonymous.4open.science/r/QC-A2S_ICLR2026-2D80/Table_6.png), that differs substantially from LLaMA in tokenizer design, attention implementation, and training methodology. The strong performance on Qwen-7B and Mistral-7B demonstrates that QC-A2S is able to transfer effectively to architectures with different design choices, supporting its broader applicability beyond the LLaMA family.

---

> ### Author Response · Authors · 2025-11-24
> **Responses to Reviewer EyDs-Part 4**
>
> [1] Yao M. et al., “Spike‑based dynamic computing with asynchronous event‑driven, sparse and dynamic neuromorphic system on chip”, Nature Communications, 2024.
>
> [2] Muir D.R., Sheik S., “The road to commercial success for neuromorphic technologies”, Nature Communications, 2025.
>
> [3] Kudithipudi, Dhireesha, et al. "Neuromorphic computing at scale." Nature 637.8047 (2025): 801-812.
>
> [4] Dalgaty, Thomas, et al. "Mosaic: in-memory computing and routing for small-world spike-based neuromorphic systems." Nature Communications 15.1 (2024): 142.
>
> [5] Yao, Man, et al. "Attention spiking neural networks." IEEE transactions on pattern analysis and machine intelligence 45.8 (2023): 9393-9410.
>
> [6] Stanojevic, Ana, et al. "High-performance deep spiking neural networks with 0.3 spikes per neuron." Nature Communications 15.1 (2024): 6793.
>
> [7] Shen, Guobin, et al. "Brain-inspired neural circuit evolution for spiking neural networks." Proceedings of the National Academy of Sciences 120.39 (2023): e2218173120.
>
> [8] Shaban, Ahmed, Sai Sukruth Bezugam, and Manan Suri. "An adaptive threshold neuron for recurrent spiking neural networks with nanodevice hardware implementation." Nature [9] Communications 12.1 (2021): 4234.
>
> [9] Chen, Xinyi, et al. "A hybrid neural coding approach for pattern recognition with spiking neural networks." IEEE transactions on pattern analysis and machine intelligence 46.5 (2023): 3064-3078.
>
> [10] Hu, Yangfan, et al. "Fast-snn: Fast spiking neural network by converting quantized ann." IEEE Transactions on Pattern Analysis and Machine Intelligence 45.12 (2023): 14546-14562.

---

> > ### Author Response · Authors · 2025-11-28
> > **Looking Forward to Further Feedback and Discussion**
> >
> > We are very grateful for your constructive review and the questions you’ve raised. We would appreciate it if you could let us know whether our revisions have addressed your concerns. We look forward to continuing the discussion and improving the manuscript based on your valuable input.

---

### Official Review · Reviewer_Rkrn · 2025-10-31

**Soundness:** 3
**Presentation:** 3
**Contribution:** 2
**Rating:** 4
**Confidence:** 4

**Summary:**

This paper addresses the issue that ANN-to-SNN conversion based on TTFS requires significant latency. The authors propose a discretization-aware framework that establishes a correspondence between discrete-TTFS-based SNNs and ANNs.

**Strengths:**

1.	The fundamental issue with existing TTFS methods requiring extremely high latency is clearly identified, which indeed limits their application on LLMs.
2.	A complete theoretical derivation from continuous to discrete TTFS neurons is provided (Theorems 1-3), establishing an equivalence relationship with quantization (Corollary 1).
3.	Evaluations are conducted across multiple LLaMA models and various datasets.

**Weaknesses:**

1.	Theorems 1 and 2 are essentially discrete versions of the continuous theory proposed by Zhao et al. (2025). The extension from continuous to discrete is relatively straightforward, with limited theoretical innovation. The error analysis in Theorem 3 lacks comparative validation between the theoretically predicted error and the actual observed error. The comparison in Figure 2 is insufficiently clear and requires more detailed explanation.
2.	The experimental setup has issues. The absence of comparisons for TTFSFormer at T=64 prevents a fair evaluation of method performance. Table 3 lacks comparisons with other models, diminishing its persuasiveness. Table 3 only investigates the impact of T, neglecting the influence of quantization bit depth. We recommend adding an analysis of the trade-off between quantization bit depth and latency.
3.	Experiments max out at 70B parameters, with no discussion of scalability for larger models.
4.	Recommendations: Add charts, visualizations, pseudocode, etc., to enhance narrative clarity.

**Questions:**

1.	The specific implementation details of the method remain somewhat ambiguous. Section 4.4 states, “we first apply established techniques, such as post-training quantization,” but does not explicitly specify which PTQ method is used. Is it PrefixQuant? If so, what distinguishes this approach from directly employing PrefixQuant?
2.	How is the trade-off between quantization bit depth n and time window T selected? What is the rationale for choosing T=64? Is this choice empirical or based on theoretical analysis?

---

> ### Author Response · Authors · 2025-11-24
> **Responses to Reviewer Rkrn-Part 1**
>
> We thank the reviewer for recognizing the clarity of our theoretical formulation, the identification of the latency bottleneck in existing TTFS methods, and the breadth of our empirical evaluation across multiple LLaMA models. We also appreciate the reviewer’s constructive comments regarding theoretical novelty, experimental completeness, and clarity of methodological description. Below, we provide detailed responses to each of the reviewer’s concerns and questions.
> >**W1**: Theorems 1 and 2 are essentially discrete versions of the continuous theory proposed by Zhao et al. (2025). The extension from continuous to discrete is relatively straightforward, with limited theoretical innovation. The error analysis in Theorem 3 lacks comparative validation between the theoretically predicted error and the actual observed error. The comparison in Figure 2 is insufficiently clear and requires more detailed explanation.
>
> We respectfully clarify that our theoretical contribution is not merely a straightforward discretization of the continuous-time formulations proposed by [1]:
> 1. **Discrete-Time Dynamics: Beyond Continuous Approximations:** Moving from continuous time to discrete time fundamentally changes the behavior of TTFS neurons because:
> (1). the spike time can no longer be solved in closed form,
> (2). the membrane dynamics become piecewise-constant,
> (3). the equivalence between spike timing and ANN activation no longer holds unless discrete constraints are explicitly enforced.
>
> 2. **Discrete-Time Error Bounds and Equivalence:** The central contribution of our work is the rigorous error analysis under discrete-time dynamics, which fundamentally advances beyond previous continuous-time formulations. Specifically, our theoretical framework provides:
> (1) show that discretization error is formally equivalent to quantization error in integer LLMs (a new theoretical link).
> (2) characterize the exact rounding-induced deviation between spike timing and ANN activations, which does not exist in continuous theory.
> (3) provide the first discrete-time error bound that is compatible with hardware implementation.
>
> These issues do not appear in any continuous-time analysis. Thus, Theorems 1 and 2 are not merely a “discrete counterpart,” but the first theoretical framework that explicitly resolves the mismatch between continuous TTFS theory and real, discrete timesteps used by hardware. Theorem 3 provides the first rigorous discrete-time error bound for TTFS conversion by precisely quantifying rounding-induced deviations in spike timing.
>
> **Comparative validation for Theorem 3**: In addition, we would like to clarify the dependence of the upper bound in Theorem 3 on the timestep $T$ and the number of layers $L$: As shown in our theoretical bound, a larger $T$ results in a smaller discretization error (Table 3 in revised paper, https://anonymous.4open.science/r/QC-A2S_ICLR2026-2D80/Table_3.png). This directly leads to improved empirical accuracy, which is consistent with our observations. Although deeper models contain more layers, the accuracy does not continue to increase. This is because the error bound in Theorem 3 accumulates with respect to the number of layers $L$ (Figure 6 in revised paper,https://anonymous.4open.science/r/QC-A2S_ICLR2026-2D80/Figure_6.png); hence, when $L$ becomes large, the accumulated discretization error offsets the benefit of increased model capacity. This explains why scaling from 7B to 13B does not yield substantial accuracy gains in our TTFS-SNN setting.
>
> Finally, to improve clarity, we have added a more detailed explanation of the algorithmic workflow in the caption of Figure 2 in revised paper (Line 123-126, https://files.imagetourl.net/uploads/1763973947917-4a57a2fa-68cb-4121-8812-948d2aef379d.png).
>
> [1] Zhao, Lusen, et al. "TTFSFormer: A TTFS-based Lossless Conversion of Spiking Transformer." Forty-second International Conference on Machine Learning.

---

> ### Author Response · Authors · 2025-11-24
> **Responses to Reviewer Rkrn-Part 2**
>
> > **W2**: The experimental setup has issues. The absence of comparisons for TTFSFormer at T=64 prevents a fair evaluation of method performance. Table 3 lacks comparisons with other models, diminishing its persuasiveness. Table 3 only investigates the impact of T, neglecting the influence of quantization bit depth. We recommend adding an analysis of the trade-off between quantization bit depth and latency.
>
> In accordance with the reviewer’s suggestions, we have further improved the experimental section as follows:
> 1. **T=64 for TTFSFormer:** To ensure a fair comparison, the revised version now includes TTFSFormer evaluated under the time window T=64, alongside our model (Table 1,https://anonymous.4open.science/r/QC-A2S_ICLR2026-2D80/Table_1.png).
> 2. **Additional LLaMA-2-13B in Table 3:** We have added LLaMA-2-13B experiments analyzing the effect of different $T$ values, similar to Table 3 (https://anonymous.4open.science/r/QC-A2S_ICLR2026-2D80/Table_3.png).
>
> **Clarification of Trade-off**: We clarify that in our conversion framework, the quantization bit depth directly corresponds to the latency (maximum time steps) in TTFS. Because our framework establishes a discrete-equivalent mapping, a higher quantization bit depth provides finer discrete spike-time resolution, enabling a larger T and thus lower discretization error, while a lower bit depth implies coarser time window, yielding a smaller T and therefore lower latency but potentially higher error.
>
>
> > **W3**: Experiments max out at 70B parameters, with no discussion of scalability for larger models.
>
> We added the experiment with LLaMA-2-70B (Table 1 and 2 in revised paper, https://anonymous.4open.science/r/QC-A2S_ICLR2026-2D80/Table_1.png and https://anonymous.4open.science/r/QC-A2S_ICLR2026-2D80/Table_2.png).
>
>
> > **W4**: Recommendations: Add charts, visualizations, pseudocode, etc., to enhance narrative clarity.
>
> In the revised paper, we have added detailed pseudocode to clarify the workflow of our method (Algorithm 1, https://anonymous.4open.science/r/QC-A2S_ICLR2026-2D80/Algorithm_1.png), and we have also included additional figures and tables to further improve the clarity and readability of the narrative (https://anonymous.4open.science/r/QC-A2S_ICLR2026-2D80).
>
> > **Q1**: The specific implementation details of the method remain somewhat ambiguous. Section 4.4 states, “we first apply established techniques, such as post-training quantization,” but does not explicitly specify which PTQ method is used. Is it PrefixQuant? If so, what distinguishes this approach from directly employing PrefixQuant?
>
> We have addressed this in the revised paper as follows:
>
> **Added detailed pseudocode:** To improve clarity, we now provide step-by-step pseudocode (Algorithm 1, https://anonymous.4open.science/r/QC-A2S_ICLR2026-2D80/Algorithm_1.png) describing the complete QC-A2S conversion pipeline, including all quantization and spiking-conversion steps.
>
> **We clarify that PrefixQuant [2] is used, motivated by our error-analysis framework:** Based on our discretization-aware theoretical analysis, PrefixQuant is adopted because it jointly controls clipping error and quantization error, enabling a quantization configuration that matches the discrete-time TTFS formulation.
>
> **Key difference from directly applying PrefixQuant: quantized neurons are replaced with TTFS spiking neurons:** Although the same quantization parameters are used, the resulting computational operators are fundamentally different: instead of performing integer arithmetic on quantized ANN neurons, we convert them into spike-based TTFS neurons with discrete kernel functions. Thus, even with identical quantization parameters, the execution model is fundamentally spiking rather than ANN-based. We also quantize nonlinear layers to ensure compatibility with spiking deployment. Unlike PrefixQuant—which focuses mainly on linear layers—we apply quantization to nonlinear operators (e.g., SwiGLU, RMSNorm, Softmax) and replace them with their corresponding spiking counterparts. This step is essential for enabling end-to-end TTFS execution. Together, these differences show that although PrefixQuant is used as a quantization backbone, **our QC-A2S pipeline is not merely PrefixQuant applied to LLMs but a principled, discretization-aware conversion framework that maps quantized LLMs into temporal coded spiking LLMs**.
>
> [2] Chen, Mengzhao, et al. "Prefixquant: Eliminating outliers by prefixed tokens for large language models quantization." arXiv preprint arXiv:2410.05265 (2024).

---

> ### Author Response · Authors · 2025-11-24
> **Responses to Reviewer Rkrn-Part 3**
>
> > **Q2**: How is the trade-off between quantization bit depth n and time window T selected? What is the rationale for choosing T=64? Is this choice empirical or based on theoretical analysis?
>
>
> We clarify that in our conversion framework, the quantization bit depth $n$ directly determines the TTFS latency (time window $T=2^n$), since each quantization level is mapped to a distinct spike time. This naturally introduces **a trade-off between latency and accuracy, which is exactly the issue our discretization-aware design aims to address**.
>
> Our initial selection of $T=64$ follows this mapping and provides a good balance between latency and performance in early experiments. To further support this choice, **we have additionally included experiments with $T=32$ and $T=16$ in the revised paper** (Table 1, https://anonymous.4open.science/r/QC-A2S_ICLR2026-2D80/Table_1.png). These results consistently demonstrate that our method maintains strong performance even under much shorter time windows, further validating the effectiveness and robustness of our framework.

---

> > ### Author Response · Authors · 2025-11-28
> > **Looking Forward to Further Feedback and Discussion**
> >
> > Thank you again for your detailed review and the insightful comments you’ve shared. We hope our responses have addressed your questions. If not, we would be grateful for any further thoughts or clarification you might have. We look forward to continuing the discussion and refining our work with your guidance.

---

### Official Review · Reviewer_uzd3 · 2025-11-03

**Soundness:** 3
**Presentation:** 3
**Contribution:** 3
**Rating:** 6
**Confidence:** 3

**Summary:**

This paper aims to convert LLM into SNN using TTFS encoding. To address the issue of high inference latency in the converted SNN, it is proven that the activation process of discrete TTFS neurons is equivalent to a quantization function. Therefore, pre-trained quantized LLM can be converted into SNN using TTFS encoding, and the conversion results are verified on the LLaMA series models.

**Strengths:**

The writing is smooth, the theory is sound, and the implementation results prove the validity of the theory.

**Weaknesses:**

1. It is still unclear what advantages this method has compared to the baseline, i.e., the quantized LLM.
2. The guiding idea of this method (i.e., converting the quantized ANN into an SNN) may not be original. Similar ideas have been proposed in previous works, and this work may only extend this idea to SNNs with TTFS encoding.[1][2]
3. The lack of a pseudocode description of the algorithm makes it insufficiently clear.

[1] Optimal ANN-SNN Conversion for High-accuracy and Ultra-low-latency Spiking Neural Networks

[2] SpikeLLM: Scaling up Spiking Neural Network to Large Language Models via Saliency-based Spiking

**Questions:**

1. What specific quantization methods are used when quantizing LLM in this paper? Are specific quantization methods required? Are both weights and activations quantized? For operations in the nonlinear parts, are the calculations performed using quantized activations and weights? Please elaborate in detail.
2. Please explain the advantages of converting the quantized LLM into an SNN? Does it improve inference efficiency, or does it reduce energy consumption? Can data support be provided?

---

> ### Author Response · Authors · 2025-11-24
> **Responses to Reviewer uzd3-Part 1**
>
> We thank the reviewer for the positive assessment of our work, including the soundness of the theory, the clarity of the writing, and the overall contribution. We appreciate the reviewer’s constructive feedback regarding the advantages over quantized LLM baselines, the originality of the conversion idea, and the clarity of algorithmic description. Below, we provide detailed responses to each of the reviewer’s comments and questions.
> > **W1**:It is still unclear what advantages this method has compared to the baseline, i.e., the quantized LLM.
>
> We thank the reviewer for the question. We would like to emphasize that the key distinction lies in the **target hardware**:
> 1. **Design for In-Memory Neuromorphic Computing:** Our SNN-based method is specifically designed as an algorithm–architecture co-design for **in-memory neuromorphic hardware**. Neuromorphic processors fundamentally differ from conventional architectures by leveraging in-memory computing, asynchronous execution, and massive parallelism, which necessitate **distinct approaches to data representation, operator execution, and computational pathways**. By exploiting event-driven computation and co-optimized memory access patterns inherent to these platforms, Spiking LLMs achieve substantial improvements in energy efficiency and latency, consistent with findings reported in prior works [1-10].
>
> 2. **Quantization Tailored to Traditional von Neumann Architectures:** Existing quantized LLM methods aim to reduce computational cost **through lower-precision computation on traditional GPUs**, yet they do not account for the compute–memory fusion and event-driven characteristics inherent to non-Von-Neumann neuromorphic processors. Consequently, conventional quantization techniques fail to exploit the architectural advantages of neuromorphic platforms. In contrast, our method leverages SNN dynamics to achieve fundamentally different efficiency gains specifically tailored to neuromorphic systems—benefits that cannot be realized by merely applying quantization to LLMs on conventional hardware.
>
> Therefore, the advantage of our method arises from its ability to fully exploit the native properties of non-Von-Neumann neuromorphic hardware—including event-driven sparsity and in-memory computing—to achieve substantial latency and energy improvements that are fundamentally unattainable through conventional quantization alone.

---

> ### Author Response · Authors · 2025-11-24
> **Responses to Reviewer uzd3-Part 2**
>
> > **W2**:The guiding idea of this method (i.e., converting the quantized ANN into an SNN) may not be original. Similar ideas have been proposed in previous works, and this work may only extend this idea to SNNs with TTFS encoding.[1][2]
> [1] Optimal ANN-SNN Conversion for High-accuracy and Ultra-low-latency Spiking Neural Networks
> [2] SpikeLLM: Scaling up Spiking Neural Network to Large Language Models via Saliency-based Spiking
>
> We sincerely thank the reviewer for the careful reading and accurate understanding of our work. As the reviewer correctly pointed out, our method indeed involves mapping representations from an ANN into an SNN. We appreciate the reviewer's clear grasp of the high-level motivation behind our approach. Our contributions lie in two fundamental aspects that have not been explored in prior TTFS-based ANN-SNN conversion:
>
> 1. **Discretization-Aware Theoretical Framework:** Existing TTFS-based conversion methods rely heavily on continuous-time assumptions, which are fundamentally misaligned with actual hardware implementations that inevitably operate in discrete time steps. Our framework is the first to:
> (1) identify the fundamental discrepancy between idealized continuous-time TTFS formulations and practical discrete-time hardware dynamics
> (2) establish a strict discrete-time correspondence between quantized Transformer activations and TTFS spiking representations;
> This discretization-aware TTFS theory did not exist in any prior conversion works and constitutes a core theoretical innovation of our paper.
>
> 2. **Discretization Error Analysis and Mitigation:** Prior methods, by neglecting discretization errors under continuous-time assumptions, require prohibitively large time steps (e.g., T = 1024–4096) to maintain acceptable approximations to floating-point activations, rendering them impractical for large language models. Building upon our discrete-time theoretical framework, we provide:
> (1) explicit characterization and rigorous analysis of conversion errors, revealing a fundamental equivalence between conversion error in TTFS representations and quantization error in fixed-point arithmetic;
> (2) effective error mitigation by directly applying post-training quantization (PTQ) techniques, leveraging the established equivalence.
> By establishing a formal correspondence between quantized LLMs and temporal spiking LLMs, our method achieves both low latency and high accuracy in spike-based large language models.
>
>
> > **W3**:The lack of a pseudocode description of the algorithm makes it insufficiently clear.
>
> We thank the reviewer for pointing this out. To address this issue, we have included a clear and concise pseudocode block in the revised paper (Algorithm 1, https://anonymous.4open.science/r/QC-A2S_ICLR2026-2D80/Algorithm_1.png) to summarize the entire QC-A2S pipeline.
>
> > **Q1**:What specific quantization methods are used when quantizing LLM in this paper? Are specific quantization methods required? Are both weights and activations quantized? For operations in the nonlinear parts, are the calculations performed using quantized activations and weights? Please elaborate in detail.
>
> Building upon our error analysis, which reveals the equivalence between conversion error and quantization error, we employ post-training quantization techniques to mitigate these errors. Specifically, we adopt PrefixQuant [1] as our quantization method, since our TTFS conversion framework requires fully static quantization to ensure deterministic spike timing. **Notably, unlike the original PrefixQuant, which targets conventional LLM deployment, we additionally quantize all nonlinear operations (e.g., softmax, layer normalization) to ensure compatibility with spike-based neuromorphic deployment**. Consequently, both weights and activations are statically quantized, and all operations—including nonlinear functions—are computed entirely in the quantized domain.
>
> [1] Chen, Mengzhao, et al. "Prefixquant: Eliminating outliers by prefixed tokens for large language models quantization." arXiv preprint arXiv:2410.05265 (2024).

---

> ### Author Response · Authors · 2025-11-24
> **Responses to Reviewer uzd3-Part 3**
>
> > **Q2**:Please explain the advantages of converting the quantized LLM into an SNN? Does it improve inference efficiency, or does it reduce energy consumption? Can data support be provided?
>
> The key advantage of converting a quantized LLM into an SNN lies in the target hardware. **Our approach is specifically designed for in-memory neuromorphic processors, which rely on event-driven computation, in-memory dataflow, and massive parallelism—properties fundamentally different from GPUs**. These architectures benefit directly from spike-based computation and have been shown in prior work [1–10] to provide substantial improvements in latency and energy efficiency. In contrast, existing quantized LLM methods are optimized for GPU-based von Neumann architectures and are generally not designed to fully exploit the compute–memory fusion or event-driven nature of neuromorphic systems, even though compute–memory integration has been explored in some inference chips and shown to substantially reduce inference power consumption. Our SNN formulation is tailored to unlock these hardware-specific advantages. At this stage, we do not have access to LLM-scale neuromorphic hardware to provide empirical energy or latency measurements, but our design aligns closely with efficiency trends consistently reported in prior neuromorphic studies.
>
>
> [1] Yao M. et al., “Spike‑based dynamic computing with asynchronous event‑driven, sparse and dynamic neuromorphic system on chip”, Nature Communications, 2024.
>
> [2] Muir D.R., Sheik S., “The road to commercial success for neuromorphic technologies”, Nature Communications, 2025.
>
> [3] Kudithipudi, Dhireesha, et al. "Neuromorphic computing at scale." Nature 637.8047 (2025): 801-812.
>
> [4] Dalgaty, Thomas, et al. "Mosaic: in-memory computing and routing for small-world spike-based neuromorphic systems." Nature Communications 15.1 (2024): 142.
>
> [5] Yao, Man, et al. "Attention spiking neural networks." IEEE transactions on pattern analysis and machine intelligence 45.8 (2023): 9393-9410.
>
> [6] Stanojevic, Ana, et al. "High-performance deep spiking neural networks with 0.3 spikes per neuron." Nature Communications 15.1 (2024): 6793.
>
> [7] Shen, Guobin, et al. "Brain-inspired neural circuit evolution for spiking neural networks." Proceedings of the National Academy of Sciences 120.39 (2023): e2218173120.
>
> [8] Shaban, Ahmed, Sai Sukruth Bezugam, and Manan Suri. "An adaptive threshold neuron for recurrent spiking neural networks with nanodevice hardware implementation." Nature [9] Communications 12.1 (2021): 4234.
>
> [9]Chen, Xinyi, et al. "A hybrid neural coding approach for pattern recognition with spiking neural networks." IEEE transactions on pattern analysis and machine intelligence 46.5 (2023): 3064-3078.
>
> [10] Hu, Yangfan, et al. "Fast-snn: Fast spiking neural network by converting quantized ann." IEEE Transactions on Pattern Analysis and Machine Intelligence 45.12 (2023): 14546-14562.

---

> > ### Author Response · Authors · 2025-11-28
> > **Looking Forward to Further Feedback and Discussion**
> >
> > We sincerely appreciate your thoughtful review and the feedback you've provided so far. We would be grateful if you could let us know whether our responses have addressed the concerns raised in your review. We look forward to continuing the discussion and refining our work with your valuable input.

---

### Author Response · Authors · 2025-11-30
**Response to Area Chair: Summary of Review Progress for Our Manuscript**

Dear Reviewers, Area Chairs, Senior Area Chairs, and Program Chairs:

We sincerely sorry for any inconvenience caused by the unforeseen events this year. We fully recognize the additional workload resulting from the recent OpenReview vulnerability and greatly appreciate the continued efforts of the project committee to uphold the integrity of the ICLR review process.

## Summary of Key Questions and Responses

To facilitate your review, we have summarized the key issues raised in the reviews and outlined the solutions we have implemented to address them:

1. **Comparison of Spiking LLMs over Quantized LLMs** (Reviewers uzd3-W1, Rkrn-Q1, EyDs-W2, M5Md-W3)

   Key distinction lies in the **target hardware**:
    * **Design for In-Memory Neuromorphic Hardware**: Our method is specifically designed to leverage in-memory computing, asynchronous execution, and parallelism, which are essential for neuromorphic processors. This approach allows significant improvements in energy efficiency and latency compared to conventional architectures.
    * **Tailored for Neuromorphic Processors**: Unlike traditional quantization methods designed for Von Neumann architectures, the proposed method uses SNN dynamics to achieve efficiency gains tailored to neuromorphic hardware. Our method exploits event-driven computation and memory fusion for more efficient performance.
2. **Absence of Comparisons for TTFSFormer at Medium Latency** (Reviewers Rkrn-W2, EyDs-W2&Q2)
    * We have added additional experiments for TTFSFormer under medium latencies, and the results show that QC-A2S consistently outperforms TTFSFormer across all latency regimes (**Tables 1 and 7**).
    * The TTFSFormer struggles with accuracy due to its need for a large time window to approximate dynamics and its failure to handle outliers during the ANN-to-SNN conversion. In contrast, the QC-A2S framework addresses these issues by stabilizing activation distributions through PrefixQuant, enabling accurate inference even under medium latencies.
3. **Energy-accuracy Trade-off** (Reviewers Rkrn-W2&Q2, EyDs-W4, M5Md)
    * The method leverages TTFS-based spiking computation, where energy consumption is primarily influenced by the number of spikes, **enabling improved accuracy without increasing energy costs when using smaller time windows**.
    * The QC-A2S framework efficiently balances accuracy and latency by coupling quantization-aware conversion, enabling higher accuracy without the need for larger time windows (**T=16**, **Tables 1 and 2**), making it more practical for real-world applications.
4. **Non-LLaMA Architectures** (Reviewer EyDs-W5&Q4)
    * To further validate this claim, we have added experiments on Qwen2-7B and Mistral-7B v0.3 (**Tables 5 and 6**). The strong performance on Qwen-7B and Mistral-7B demonstrates that QC-A2S is able to transfer effectively to architectures with different design choices, supporting its broader applicability beyond the LLaMA family.
    * QC-A2S is not tied to the LLaMA architecture. Our conversion pipeline operates directly at the level of quantized linear layers and attention modules, rather than relying on architecture-specific heuristics.
5. **Theoretical Innovation and Practical Guidance Significance** (Reviewers uzd3-W2, Rkrn-W1, EyDs-W1&Q1)
    * Innovation:  The theoretical contribution of this work goes beyond a straightforward discretization of continuous-time formulations. It provides a **rigorous error analysis in discrete-time dynamics**, introducing **the first discrete-time error bounds and equivalences for TTFS conversion**.
    * Guidance: The error bound in Theorem 3 depends on the timestep T and the number of layers L. While larger T reduces discretization error and improves accuracy, increasing the number of layers L leads to accumulated discretization error that offsets the accuracy gains, explaining why scaling from 7B to 13B layers does not result in significant accuracy improvements (**Table 3, Figure 6**).

## Main Contribution of Our Paper
To **mitigates the high latency caused by continuous-time assumptions in TTFS-based A2S**, our paper introduces the QC-A2S conversion, grounded in a **discretization-aware theoretical framework**, achieving **latency-efficient** temporal-coding spiking LLMs.

We have addressed all the reviewers’ comments in our response and prepared a revised paper. **Within the limited time available, all reviewers have not yet responded**, we respectfully request that the AC, SAC, and PC consider any score improvements warranted by our responses.

We express our sincere respect for your extra contributions!

Best Regards,

The Authors

---

### Meta-Review · Area_Chair_opPj · 2026-01-06

**Summary:**

This paper proposes QC-A2S, a discretization-aware ANN-to-SNN conversion framework for time-to-first-spike (TTFS) coded spiking LLMs. By establishing a theoretical correspondence between discrete-time TTFS neurons and quantized ANN activations, the method aims to significantly reduce inference latency compared to prior continuous-time TTFS approaches.

**Reviewer Concerns:**

Reviewer concerns mainly center on the practical significance and novelty of the proposed approach relative to existing quantized LLMs and prior ANN-to-SNN conversion methods.

**Reviewer Scores:**

One reviewer rated the paper marginally above the acceptance threshold and acknowledged the sound theory and strong experimental validation, while several others rated the work as marginal reject, primarily due to concerns about novelty, baseline comparisons, and the absence of hardware-level validation. As a result, despite generally positive views on technical correctness, the reviews did not converge toward a clear positive consensus.

---

### Decision · Program_Chairs · 2026-01-26

Reject